# Quantitative proteomics analysis reveals important roles of N-glycosylation on ER quality control system for development and pathogenesis in *Magnaporthe oryzae*

**Xiao-Lin Chen**[1,2], **Caiyun Liu**[1], **Bozeng Tang**[3], **Zhiyong Ren**[1], **Guo-Liang Wang**[2,4], **Wende Liu**[2]*

**1** The Provincial Key Laboratory of Plant Pathology of Hubei Province, College of Plant Science and Technology, Huazhong Agricultural University, Wuhan, China, **2** State Key Laboratory for Biology of Plant Diseases and Insect Pests, Institute of Plant Protection, Chinese Academy of Agricultural Sciences, Beijing, China, **3** The Sainsbury Laboratory, University of East Anglia, Norwich, United Kingdom, **4** Department of Plant Pathology, Ohio State University, Columbus, Ohio, United States of America

* wendeliu@126.com

**Data Availability Statement:** Proteomic data presented in this paper are avalabel in the following PRIDE file- ProteomeXchange title: N-

## Abstract

Genetic studies have shown essential functions of N-glycosylation during infection of the plant pathogenic fungi, however, systematic roles of N-glycosylation in fungi is still largely unknown. Biological analysis demonstrated N-glycosylated proteins were widely present at different development stages of *Magnaporthe oryzae* and especially increased in the appressorium and invasive hyphae. A large-scale quantitative proteomics analysis was then performed to explore the roles of N-glycosylation in *M. oryzae*. A total of 559 N-glycosites from 355 proteins were identified and quantified at different developmental stages. Functional classification to the N-glycosylated proteins revealed N-glycosylation can coordinate different cellular processes for mycelial growth, conidium formation, and appressorium formation. N-glycosylation can also modify key components in N-glycosylation, O-glycosylation and GPI anchor pathways, indicating intimate crosstalk between these pathways. Interestingly, we found nearly all key components of the endoplasmic reticulum quality control (ERQC) system were highly N-glycosylated in conidium and appressorium. Phenotypic analyses to the gene deletion mutants revealed four ERQC components, Gls1, Gls2, GTB1 and Cnx1, are important for mycelial growth, conidiation, and invasive hyphal growth in host cells. Subsequently, we identified the Gls1 N-glycosite N497 was important for invasive hyphal growth and partially required for conidiation, but didn't affect colony growth. Mutation of N497 resulted in reduction of Gls1 in protein level, and localization from ER into the vacuole, suggesting N497 is important for protein stability of Gls1. Our study showed a snapshot of the N-glycosylation landscape in plant pathogenic fungi, indicating functions of this modification in cellular processes, developments and pathogenesis.

glycoproteome of Magnaporthe oryzae; ProteomeXchange accession: PXD015522; Project Webpage: http://www.ebi.ac.uk/pride/archive/projects/PXD015522. All other relevant data are within the manuscript and its Supporting Information files.

**Funding:** WDL was supported by the National Programme for Support of Top-notch Young Professionals. This work was funded by the National Natural Science Foundation of China (31422045) to W.D.L, and the Open Research Fund of State Key Laboratory for Biology of Plant Diseases and Insect Pests (SKLOF201905) to X.-L. C. The funders had no role in study design, data collection and analysis, decision to publish, or preparation of the manuscript.

**Competing interests:** The authors have declared that no competing interests exist.

## Author summary

The fungal pathogen *Magnaporthe oryzae* can cause rice blast and wheat blast diseases, which threatens worldwide food production. During infection, *M. oryzae* follows a sequence of distinct developmental stages adapted to survival and invasion of the host environment. *M. oryzae* attaches onto the host by the conidium, and then develops an appressorium to breach the host cuticle. After penetrating, it forms invasive hyphae to quickly spread in the host cells. Numerous genetic studies have focused on the mechanisms underlying each step in the infection process, but systemic approaches are needed for a broader, integrated understanding of regulatory events during *M. oryzae* pathogenesis. Many infection-related signaling events are regulated through post-translational protein modifications within the pathogen. N-linked glycosylation, in which a glycan moiety is added to the amide group of an asparagine residue, is an abundant modification known to be essential for *M. oryzae* infection. In this study, we employed a quantitative proteomics analysis to unravel the overall regulatory mechanisms of N-glycosylation at different developmental stages of *M. oryzae*. We detected changes in N-glycosylation levels at 559 glycosylated residues (N-glycosites) in 355 proteins during different stages, and determined that the ER quality control system is elaborately regulated by N-glycosylation. The insights gained will help us to better understand the regulatory mechanisms of infection in pathogenic fungi. These findings may be also important for developing novel strategies for fungal disease control.

## Introduction

Asparagine-linked protein glycosylation, or N-glycosylation, is one of the most complex and abundant post-translational modifications of eukaryotic proteins [1]. N-glycosylation can change the folding, stability, quality control, sorting, and localization of target proteins [2]. By altering protein functions, N-glycosylation mediates diverse biological processes, including differentiation, growth, development, and intercellular communication [3–4]. As an enzymatically catalyzed process, N-glycosylation occurs in the endoplasmic reticulum (ER). This process includes assembling the glycans on a lipid carrier dolichol pyrophosphate (Dol-PP) in the ER membrane and transferring the glycans to target proteins [5]. During the latter process, the assembled oligosaccharide is transferred *en bloc* from the lipid carrier Dol-PP to the protein substrate, which occurs on select asparagine glycosylation sequons (N-X-S/T; X≠P) as soon as the protein substrate arrives in the ER lumen [6]. The N-linked glycan structure affects protein folding and facilitates the ER quality control system in identifying properly folded proteins. In the secretory pathway, N-glycan linked proteins are transported to the Golgi apparatus, where more complex, hybrid and paucimannose-type N-glycans are added into the N-glycan structures to form mature N-glycosylated proteins [7–8]. If an N-glycan linked protein is not properly folded, it will be recycled in the ER-associated degradation pathway [9].

Over the past decade, the biological functions of protein N-glycosylation have been analyzed in fungi including the human pathogen *Candida albicans* [10–12] and the plant pathogens *Ustilago maydis*, *Magnaporthe oryzae* and *Mycosphaerella graminicola* [13–16], establishing that N-glycosylation is essential for evasion of host immunity or establishment of infection of fungi. However, it is still largely unknown why N-glycosylation is important for fungal pathogenesis. Current understanding of N-glycosylation has largely been established through functional genetic studies of the N-glycosylation pathway, while the N-glycosylated target proteins executing biological functions have received far less attention. Therefore,

systematic studies of the targets of N-glycosylation, including profiling of the N-glycoproteins and N-glycosites, will be critical to understand the mechanistic role of this modification in the infection process. N-glycoproteomes of several model organisms have been performed and investigated in the past decade, including humans and plants [17–21]. However, few N-glycoproteome studies focused on the global functions of N-glycosylation during fungal pathogenesis.

*Magnaporthe oryzae* is the causal agent of rice blast disease, one of the most destructive rice diseases in the world [22]. *M. oryzae* begins the infection process when a conidium touches the surface of a host leaf. The conidium then germinates and develops into a dome-like appressorium [23–24]. The appressorium generates high turgor pressure by accumulating compatible solutes to facilitate the penetration of the plant surface. After penetration, the fungus initiates a biotrophic growth stage for a short period, then switches into a necrotrophic growth stage and produces asexual conidia to disseminate. In this study, we used label-free quantification of N-glycosylation sites at different infection stages to identify the functions of N-glycosylation during the development and infection of *M. oryzae*. A total of 559 unique N-glycosite-containing peptides from 355 N-glycoproteins were identified and quantified. We found that N-glycosylation prominently modifies not only the cell wall and plasma membrane proteins, but also a large number of proteins from the secretory pathway with crucial functions in protein glycosylation, folding, quality control and secretion. N-glycosylation coordinated proteins for mycelial growth, conidium formation, and appressorium formation. We demonstrated that N-glycosylation can regulate endoplasmic reticulum quality control (ERQC) factors to facilitate penetration and host infection. The results of this study provide an overview and shed new light on the N-glycosylation protein mediated regulatory functions, which can help to unravel the relationship of N-glycosylation and pathogenesis in *M. oryzae*.

## Results

### Biological importance of N-glycosylation at different development stages of *M. oryzae*

We firstly analyzed the changes in global N-glycosylation levels for the mycelium, conidium and appressorium stages. The invasive hyphae (IH) stage was omitted due to difficulty separating the IH from host tissue. Western blot analysis using horseradish peroxidase-conjugated concanavalin A (ConA-HRP) as an antibody revealed that the N-glycosylated proteins of the appressoria and conidia were more abundant than that of the mycelia (Fig 1A). We also performed gene expression profiling of the N-glycosylation pathway. Results showed that expression levels of the late pathway genes, which form complex-, hybrid- and paucimannose-type N-glycans in the Golgi apparatus, were elevated in the conidia and early appressorium stage (S1 Fig). These results suggested that the strength of N-glycosylation increased by adding more N-glycan chains during conidium and appressorium formation.

The existence and localization of N-glycoproteins was assessed in the mycelia, conidia, appressoria and invasive hyphae (IH) of *M. oryzae* using the FITC fluorescence-fused lectin concanavalin A (ConA-FITC) staining assay. ConA can recognize terminal α-D-mannose and α-D-glucose residues, which make up the main N-glycan structure [25]. Fluorescence was detected in the outer cell layer of all ConA-FITC stained tissues (Fig 1B), indirectly suggesting that N-glycosylated proteins were present in all developmental stages of *M. oryzae*. Compared with that in the mycelium, fluorescence was much stronger in the conidium, appressorium and invasive hyphal growth stages, indicating increased N-glycosylation during these development stages. We also specifically stained the GPI-anchored glycoproteins (the main glycoproteins in cell wall) with a fluorescently labelled aerolysin (FLAER) dye (Alexa Fluor 488

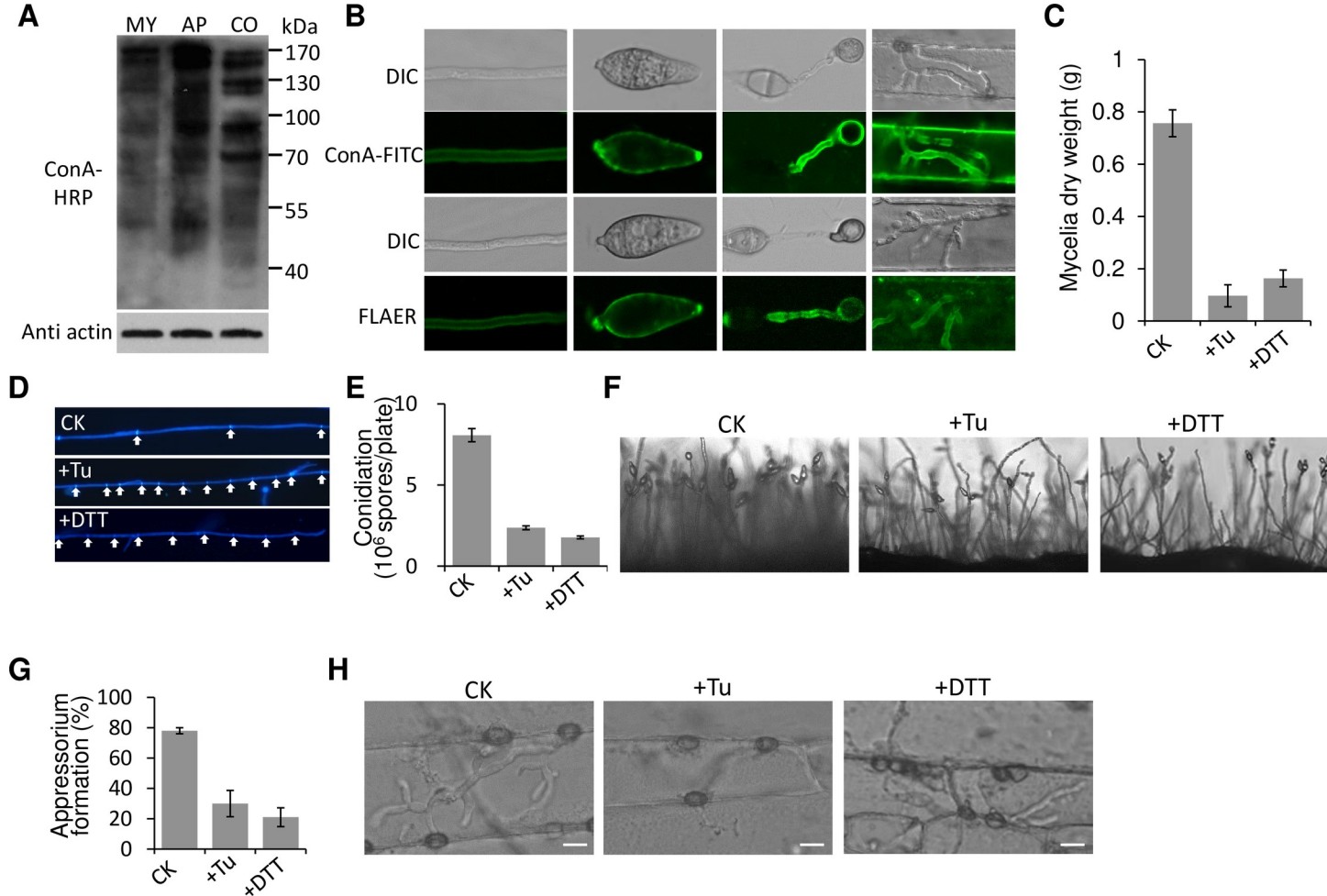

**Fig 1. Biological importance of N-glycosylation.** (A) Different N-glycosylated protein levels tested by Western blot using the ConA-HRP antibody. MY, mycelia; AP, appressoria; CO, conidia. (B) ConA-FITC and FLAER staining assay in different tissues of *M. oryzae*. Mycelia, conidia, appressoria and invasive hyphae were stained with 10 μg/mL ConA-FITC or 50 nM FLAER. ConA-FITC, FITC fluorescence-fused lectin concanavalin A; FLAER, fluorochrome (Alexa 488)–labeled inactivated aerolysin. Bar, 10 μm. Effect of 5 μg/mL tunicamycin (Tu) or 10 mM dithiothreitol (DTT) on: (C) mycelia dry weight, (D) the length of mycelial cell, (E) conidiation, (F) conidiophore formation, (G) appressorium formation ratio, (H) invasive hypha formation. Bar, 10 μm.

proaerolysin). The FLAER can selectively bind to GPI-anchored glycoproteins on the anchor [26], which is independent of N-glycan [27] and can exclude interference of carbohydrates and glycolipids. The result indicated that the FLAER staining showed a similar fluorescence level changes as detected by the ConA-FITC (Fig 1B). Taken together, these results were consistent with that of the western blot analysis (Fig 1A).

Tunicamycin, an N-glycosylation inhibitor that blocks the formation of N-glycosidic protein-carbohydrate linkages [28], was used to determine whether N-glycosylation is important for *M. oryzae* development and pathogenesis. We found that 5 μg/mL tunicamycin significantly inhibited mycelial growth, mycelial cell length, conidiation and conidiophore formation (Fig 1C–1F). More importantly, for infection, tunicamycin suppressed appressorium formation, reducing the appressorium formation rate from 80% in non-treated conidia to 30% in treated conidia (Fig 1G). To establish whether tunicamycin could block invasive hyphal growth within host cells, we added 5 μg/mL tunicamycin to conidia droplets on barley leaves at 18 dpi, when the fungal appressoria were beginning to form penetration pegs. At 24 hpi,

fewer branches had formed in the IH treated with tunicamycin than in the untreated strain (60%, Fig 1H), demonstrating that tunicamycin can prevent invasive hyphal growth in host cells. Because tunicamycin (TM) and dithiothreitol (DTT) are two well characterized ER stress inducers, we also used low concentration of DTT (10 mM) to test its effect on *M. oryzae* development. Obviously, similar phenotypic defects of the wild-type strain were observed comparing with the tunicamycin treatment (Fig 1C–1H). These data indicated that N-glycosylation is therefore crucial for all tested development and infection stages, probably by partially affecting process of the ER stress response.

## Quantitative N-glycoproteomics profiling of different development stages in *M. oryzae*

Label-free quantitative N-glycoproteomics was used to identify changes in the N-glycosylation state of target proteins during *M. oryzae* mycelium, conidium, and appressorium formation. Sampling points corresponded to five stages of fungal development (Fig 2A). At 6 h after conidial inoculation onto a hydrophobic surface, germ tube tips swelled and cellular components migrated into the nascent appressorium. At 12 hpi, the appressorium became pigmented with melanin, and at 24 hpi the appressorium had matured and began to penetrate the surface.

Total protein was collected from mycelia and conidia germinated on the hydrophobic surface at 0, 6, 12 and 24 h to identify N-glycosylation changes associated with developmental stage. The experimental strategy is depicted in Fig 2B. Total protein collected from two biological replicates of each sample was digested with trypsin and subjected to glycopeptide enrichment by lectin affinity chromatography. Efficiency of N-glycoprotein enrichment was enhanced by combining three widely used lectins: Concanavalin A (ConA), Wheat Germ Agglutinin (WGA) and *Ricinus communis* Agglutinin (RCA120). The enriched peptides were treated with immobilized trypsin and PNGase-F in $^{16}$O or $^{18}$O. During this process, the N-glycosylation site was labeled with one atom of $^{16}$O or $^{18}$O and the C terminus of the peptide was labeled with two $^{16}$O or $^{18}$O. When $^{16}$O and $^{18}$O-labeled samples were mixed at a 1:1 ratio, the labeled glycopeptides could be detected by a mass shift of 6 Da using Q-Exactive LC-MS/MS, while a 4 Da mass shift represented the $^{16}$O or $^{18}$O-labeled non-glycopeptides. Analyzing the concentration ratio yielded the quantity of the N-glycosylated peptides. Only peptides identified with a PeptideProphet probability score $\geq$ 0.95 ($\sim$1.2% error rate) were retained for analysis. The resulting data were validated against a *M. oryzae* proteome database (http://fungi.ensembl.org/Magnaporthe_oryzae/Info/Index) for peptide/protein identification. The mass spectrometry proteomics data have been deposited to the ProteomeXchange Consortium [29] via the PRIDE [30] partner repository with the dataset identifier PXD015522.

We identified 559 N-glycosylation sites from a total of 355 proteins in all tissues (S1 Table and S2 Table). Of these, 313, 324 and 274 N-glycosylation sites were identified in the mycelium, conidium and appressorium, respectively, with 156 sites common to all tissues (Fig 2C). We found that 204 sites were identified in both the mycelium and conidium, 222 sites in both the conidium and appressorium and 179 sites in both the mycelium and appressorium. There were 86, 54 and 29 N-glycosylation sites that were identified specifically in the mycelium, conidium and appressorium, respectively.

## Sequence features of N-glycoproteome

Among the 355 total N-glycosylated proteins, 59.7% had only one N-glycosite. Proteins with two, three and four N-glycosites accounted for 20.6%, 9.9% and 4.2% of the total N-glycosylated proteins, respectively (Fig 2D). Notably, 12 proteins (3.4%) were N-glycosylated at six or

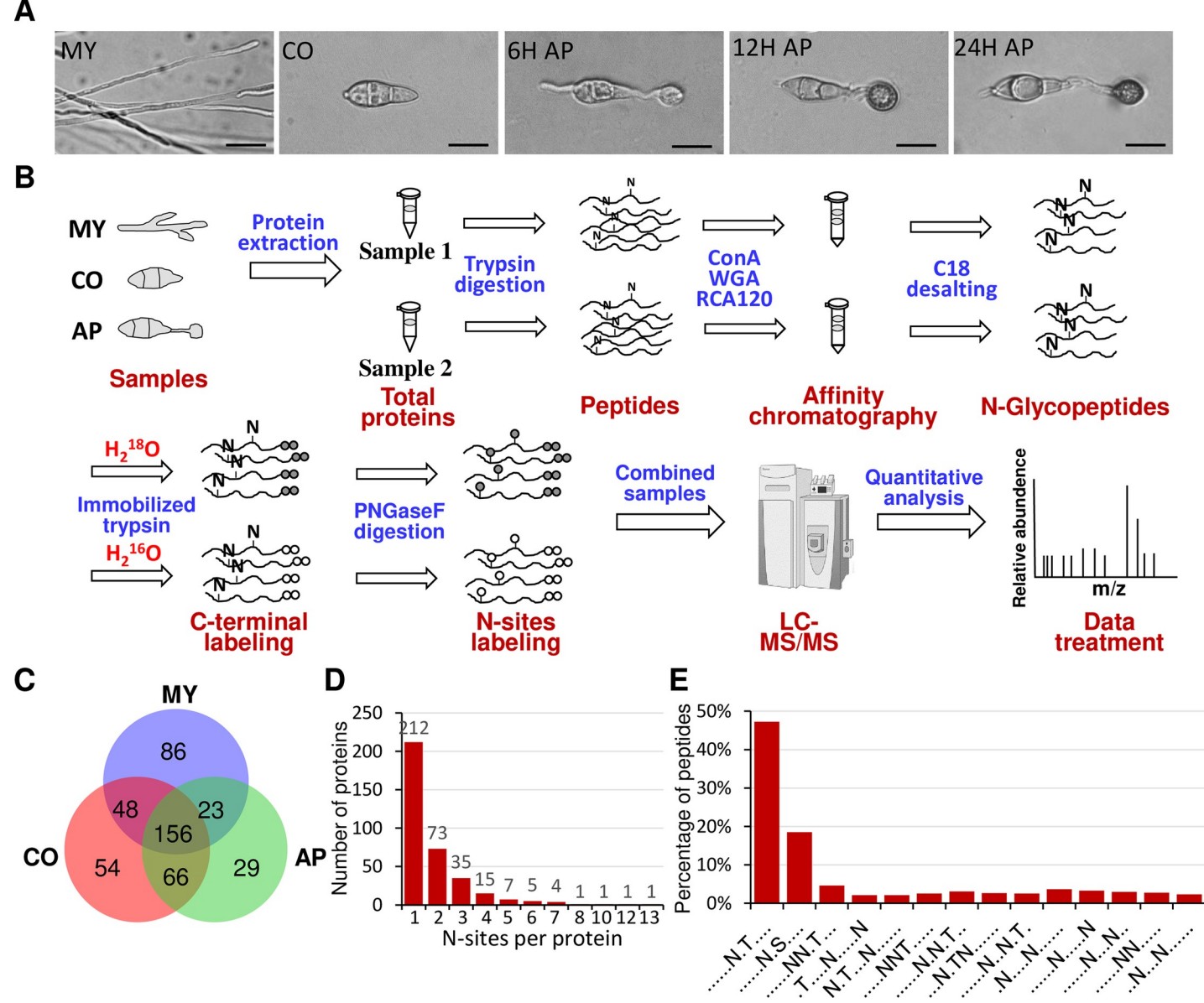

**Fig 2. Schematic for quantitative analysis strategy of the N-glycoproteome in *M. oryzae*.** (A) Light micrographs of *M. oryzae* mycelia, conidia, and conidia germinated on a hydrophobic surface showing the development of appressoria at the time points used in the N-glycoproteomics study. (B) Flow chart of the integrated strategy for quantitative analysis of the N-glycoproteome. Samples were digested with trypsin after removal of high-abundance proteins. Glycopeptides were enriched with lectin chromatography from two different samples, desalted, and then catalyzed by immobilized trypsin and PNGase-F in $^{18}O$ or $^{16}O$ water, as indicated. Equal amounts of $^{16}O$ and $^{18}O$ -labeled glycopeptides were mixed, and the 6 Da mass shifts were generated between paired, labeled glycopeptides, which could be identified by subsequent LC-MS/MS. (C) Venn diagram showing the number of N-proteins identified in different stages. (D) Distribution of the number of glycosylated sites on proteins. (E) Relative frequency plots of the N-glycosylation sequon (N-X-S/T) in the entire population of N-glycopeptides. MY, mycelia; CO, conidia; AP, appressoria.

more sites (S1 Table), including levanase MGG_05785 (13 sites), trehalase MGG_01261 (12 sites) and glycoside hydrolase MGG_13429 (10 sites).

We used a motif-X algorithm software to generate sequence logos [31]. Probable amino acid positions surrounding N-glycosylation sites were computed to identify specific sequence motifs surrounding asparagine residues in identified N-glycosites. An oligosaccharide chain is commonly attached to asparagine (N) occurring in the tripeptide sequence N-X-T or N-X-S

(X represents any amino acid except Pro). As expected, the majority of the N-oligosaccharide chains were attached to N-X-T (47.2%) and N-X-S (18.5%), although other motifs accounted for an over a third of all N-glycosylation sites (Fig 2E).

## Classification of identified N-glycoproteins

The BLAST2GO application GO (Gene Ontology) functional classification analysis was used with all identified N-glycosylated proteins to understand the potential regulatory roles (biological, molecular and cellular) of N-glycosylation in *M. oryzae* (Fig 3A). The largest number of N-glycosylated proteins were linked to processes of carbohydrate metabolism. Numerous N-glycosylation sites were also observed on proteins with predicted involvement in the oxidation-reduction reactions or the metabolism of nitrogen or lipid compounds, suggesting that N-glycosylation is important for regulating the metabolism of various nutrients and maintaining redox balance. Diverse other biological processes were also predicted to be targeted by N-glycosylation, including localization of cellular components, response to stimuli, and host interactions (Fig 3A).

For the molecular functions, the majority of N-glycosylated sites were found in various predicted protein hydrolases and peptidases, suggesting that N-glycosylation may regulate protein degradation and amino acid cycling. Proteins with predicted oxidoreductase and transferase activities were also N-glycosylated (Fig 3A). Cellular component predictions placed most of the N-glycosylated proteins in the cytoplasm, membrane-bounded organelles and ER (Fig 3A). These results are consistent with the understanding that N-glycosylation primarily occurs in the cytoplasm, membrane-bounded organelles and secretory system.

N-glycosylated proteins were analyzed for predicted biochemical pathway involvement using KEGG (Kyoto Encyclopedia of Genes and Genomes) analysis. Results showed that proteins involved in the ER protein secretion system were significantly enriched (Fig 3B). Other pathways enriched in N-glycosylated proteins included N-glycan biosynthesis, autophagy, synthesis of GPI-anchors, and metabolism of starch, sugars, glycerolipids, and glyoxylate (Fig 3B). Together, GO annotation and KEGG pathway analysis indicate that N-glycosylation may regulate diverse cellular processes, including the metabolism of carbohydrates, proteins and lipids, and the processing, localization and secretion of secretory proteins. Many of the N-glycosylated proteins identified also have important roles in the developmental biology of *M. oryzae* as discussed below.

## N-glycoprotein protein interaction networks

We generated an interaction network using the Search Tool for Retrieval of Interacting Genes/Proteins (STRING) database to further understand the significance and extent of N-glycosylation in *M. oryzae*. A number of sub-networks were identified, including glycan biogenesis, ER secretory pathway, carbohydrate metabolism and spore cell wall biogenesis (Fig 3C). The interaction network having the highest number of N-glycosylated proteins is involved in glycan biogenesis, which includes proteins involved in N-glycosylation (STT3, OST1, WBP1, MNN2, MNN9, MNN10), O-glycosylation (PMT2, PMT4) and protein folding and quality control (GLS2, RTB1, CNE1, KRE5), indicating that N-glycosylation plays key roles in the glycan biogenesis system itself. We also observed that N-glycosylated proteins in the ER secretory pathway were enriched. Two interaction networks involved in carbohydrate metabolism and spore cell wall biogenesis, respectively, contained at least six N-glycosylated proteins (Fig 3C).

## Comparing N-glycoproteins at different developmental stages of *M. oryzae*

We sought to understand the N-glycosylation protein dynamics of the different developmental stages of *M. oryzae* by identifying N-glycosites that are unique to some stages or that change in

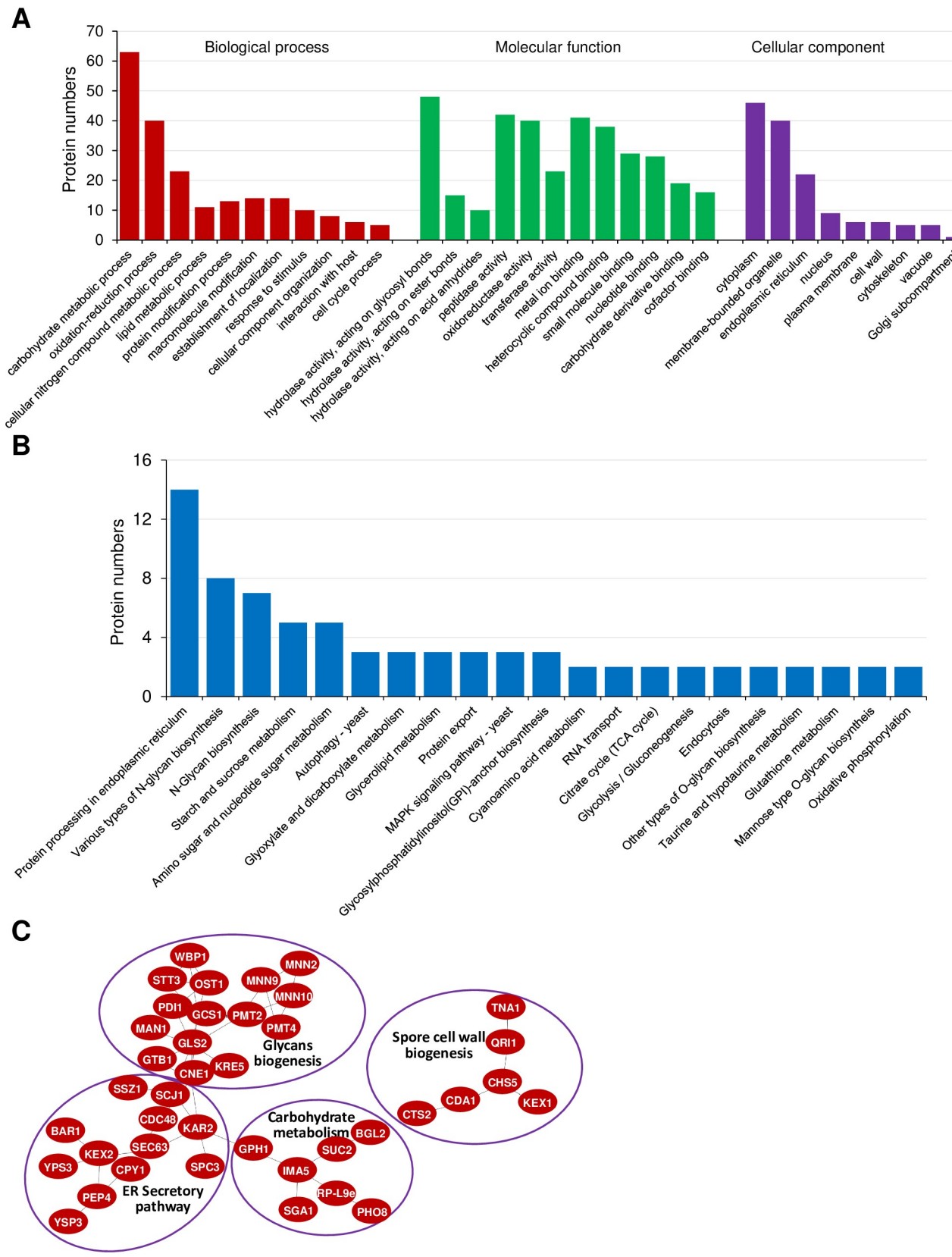

**Fig 3. Classification of identified and protein interactions of the N-glycoproteome.** (A) Functional classification of identified N-glycoproteins based on Gene Ontology analysis. (B) KEGG pathway enrichment of the N-glycosylated proteins. (C) Protein interaction networks of the N-glycoproteins. Protein interaction networks were generated with the complete list of N-glycosylated proteins using the STRING database and visualized using the Cytoscape program.

abundance during development. First, we performed principle component analysis (PCA) to the N-glycoproteome of different samples, including mycelia (MY), conidia (CO), 6h appressoria (AP), 12h AP, and 24h AP. The result demonstrated that both replicates exhibited similar expression pattern for each sample, indicated well repetitiveness. The PCA result also suggested that the MY, CO and AP samples were different in N-glycoproteins expression patterns from each other, while the 6h AP, 12h AP, and 24h AP samples were similar to each other (Fig 4A). Subsequent Pearson correlations and clustering analyses to the different samples also indicated 6 h AP, 12 h AP, and 24 h AP samples enrich a similar abundance of N-glycoproteins, but different from the MY or CO samples (Fig 4B and 4C).

Therefore, we combined the average N-glycosylation levels of N-glycosites at the 6h AP, 12h AP, and 24h AP formation stages. This resulted in three sets of N-glycosites corresponding to the highest N-glycosylation levels in the mycelia, conidia and appressoria (S3 Table): 212 sites were highly N-glycosylated in mycelia, 254 in conidia, and 261 in appressoria (Fig 2C). A total of 486 sites on 278 proteins were differentially N-glycosylated in different developmental stages (S3 Table). One hundred thirty-one proteins of diverse predicted functions were highly N-glycosylated in all tested tissues (S4 Table). Conversely, some proteins showed a clear development-specific pattern of N-glycosylation levels. Transient increases and decreases in N-glycosylation levels of these proteins might shed more light on the function of several basic biological processes during the differentiation process. The biological impact of the observed N-glycosylation mediated function is addressed in the following sections.

## N-glycosylation is involved in different cellular processes during development

The strength and rigidity of the fungal cell wall can be adjusted to facilitate growth and development. For the pathogenic fungi, the cell wall is vital for adhesion, penetration and invasive hyphal growth during infection. Our results identified numerous N-glycosylated cell wall-related proteins, including cell wall chitin transglycosylases UTR2 and CRH1 (transfer chitin to beta(1–6) and beta(1–3) glucans), endo-beta-1,3-glucanase BGL2 (incorporate newly synthesized mannoprotein molecules into the cell wall), β-1,3-glucanosyltransferases Gel3 and Gel4 (the GPI-anchored membrane proteins required for cell wall biosynthesis), mannan endo-1,6-alpha-mannosidase DCW1 and chitin synthase CHS6 (Fig 5A). Some cell wall proteins, such as DCW1, ECM14, Mad1 and ECM33, had noticeably increased N-glycosylation levels in the mycelia (Fig 5A). Others, including fasciclin-like protein MoFLP1 [32], flocculin FLO11 [33], chitin synthase genes CHS5 and CHS6 [34], chitin-binding protein CBP1 [35], ECM38 and Gel3 [36], were highly N-glycosylated in the conidia and appressoria (Fig 5A). Chitin is a major component of fungal cell wall and is synthesized by chitin synthases (Chs), including CHS5 and CHS6, which was found to be important for plant infection in maintaining polarized growth in vegetative and invasive hyphae [34]. MoFLP1 was reported to encode a fungal fasciclin-like protein, which is important for conidiation and pathogenicity in *M. oryzae* [32]. CBP1 encodes a putative extracellular chitin-binding protein, which plays an important role in the hydrophobic surface sensing during appressorium differentiation [35]. These results show that *M. oryzae* uses N-glycosylation to modify different cell wall proteins and structures when coordinating distinct developmental processes.

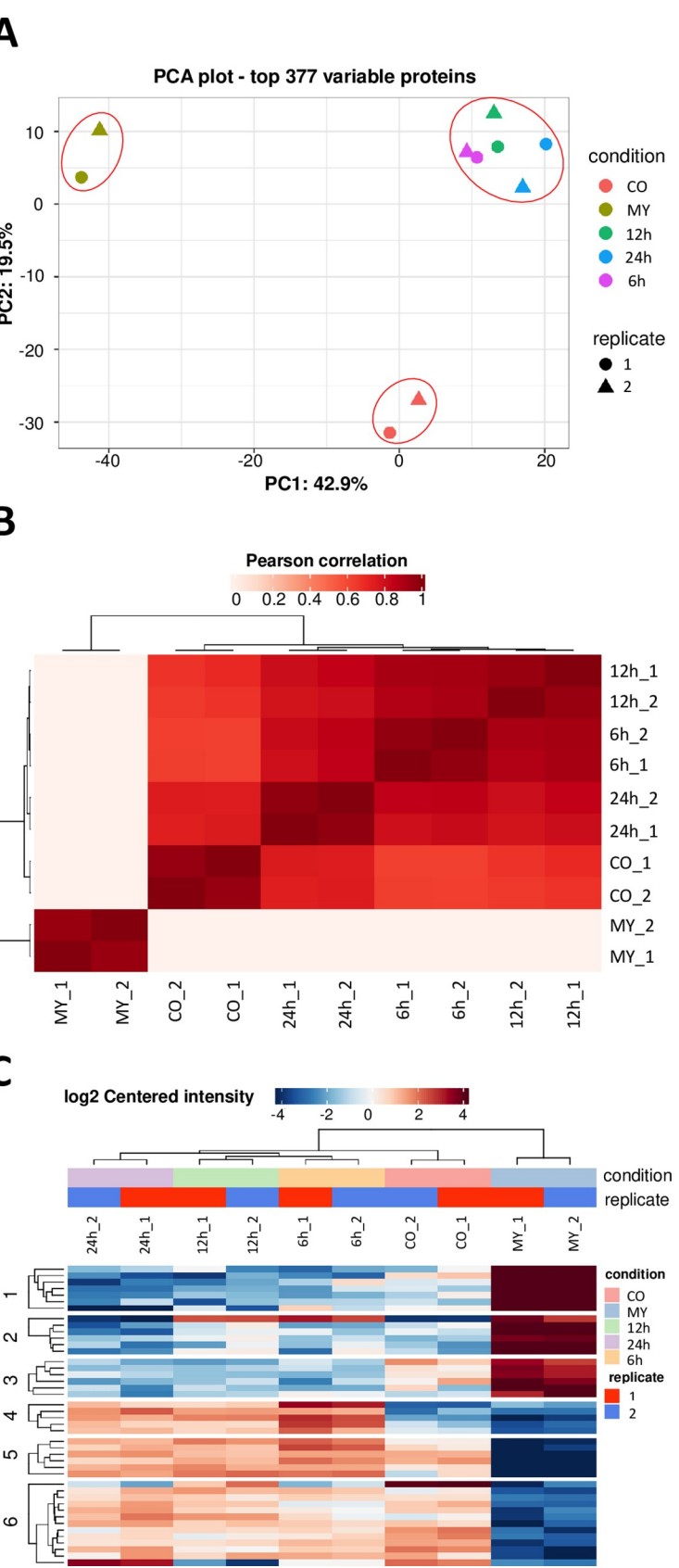

**Fig 4. Differentiation of the N-glycoproteome among different samples.** (A) The principal component analysis (PCA) was carried out to get an overview of the data to detect batch effects, and assess differences between replicates. After filtering the missing values, the retained proteins were identified in two replicates of at least one condition. (B) The heatmap is plotted to visualized the Pearson correlations between the different samples. Colors indicate no correlation (white) and strong correlation (red). (C) The heatmap shows a clustering analysis of replicates, and indicates 6 h, 12 h, and 24 h appressoria enrich a similar abundance of proteins. Significant expression differences can be compared between individual samples during the differentiation process. The rows represent different proteins and are clustered in six groups by k-means clustering. Rows and columns are hierarchically clustered on Euclidean distance. Colors represent enrichment in the linkage versus control (red: enriched; blue: depleted). MY, mycelia; CO, conidia; 6h, 12h, 24h, appressoria (AP) at 6, 12 and 24 hpi.

Fungi need to activate various cellular processes to assimilate nutrients and facilitate vegetative growth. Some proteins related to nutrient assimilation, such as proteases and peptidases, cell wall, lipid metabolism, and cellular redox homeostasis related proteins, were highly N-glycosylated in the mycelia (Fig 5B and S4 Table). Collectively, N-glycosylated proteins in the mycelia were mainly involved in cell wall biogenesis and nutrient utilization.

In conidia, the storage reserves, including glycogen, trehalose and lipid bodies, are accumulated during the conidium formation, and is required for the function of appressorium. The conidium storage reserves are presumably used to fuel biosynthetic processes and turgor generation in the developing appressorium. Gph1 is a glycogen phosphorylase that is involved in the breakdown of glycogen and is required to mobilize glycogen and full virulence [37]. One glycosylation site, N716, was detected on Gph1 (S1 Table). TRE1, which encodes an enzyme localized in the cell wall with characteristics of both neutral and acidic trehalases [38],

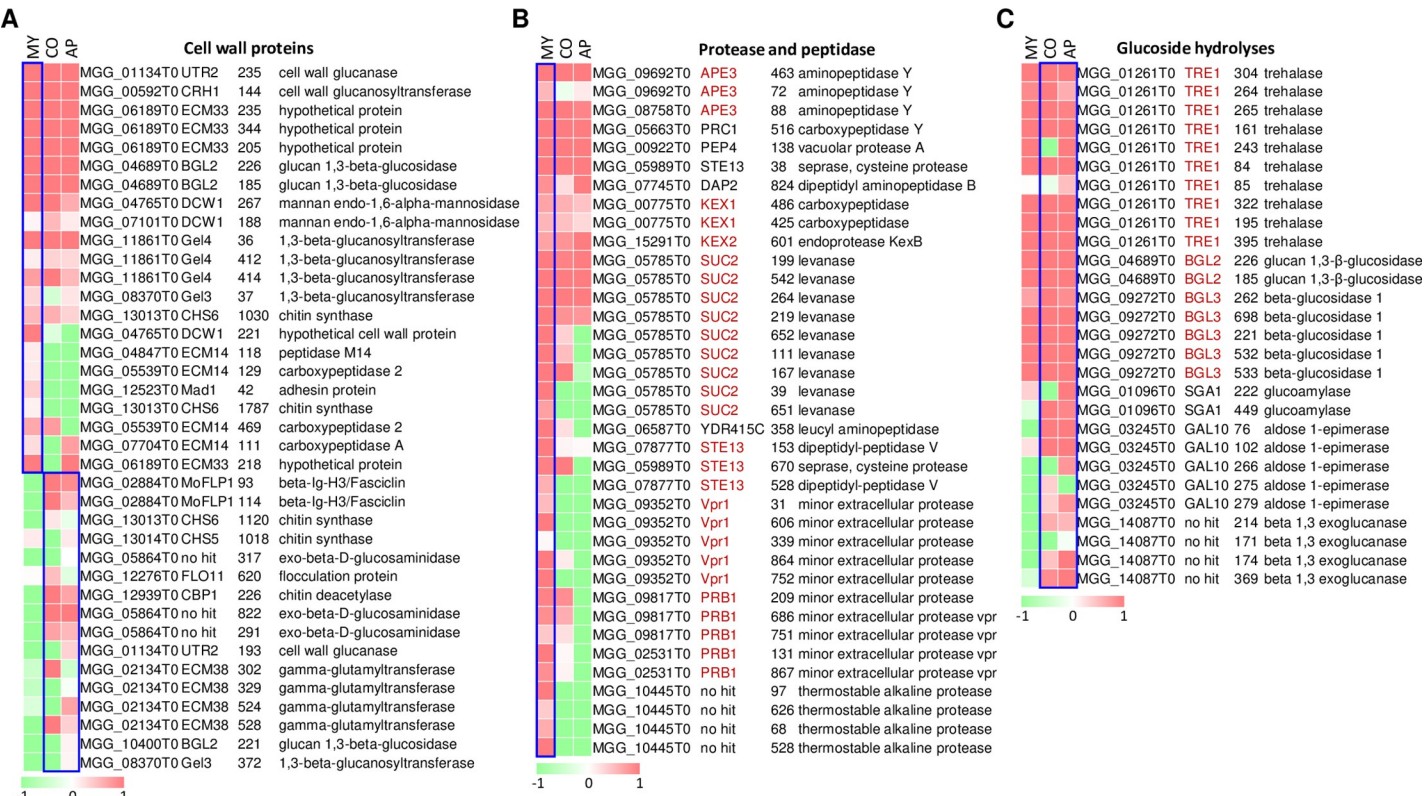

**Fig 5. Different cellular processes mediated by N-glycoproteins.** The heat map shows the N-glycosylation profile of various N-glycoproteins in different developmental stages, including (A) cell wall proteins, (B) protease and peptidase, and (C) glycoside hydrolyses. Data is the mean of two biological replicates. The colored bar represents the scale for the log10 fold change in expression from green (-2) to red (2), white is set as 0. MY, mycelia; CO, conidia; AP, appressoria.

contained 10 N-glycosites, most of which had elevated N-glycosylation levels in the conidia (Fig 5C). Glucan 1,3-beta-glucosidases BGL2 and BGL3 and UDP-glucose-4-epimerase GAL10 are involved in glucose metabolism and important for accumulating conidium storages. BGL2, BGL3 and GAL10 had elevated N-glycosylation levels in the conidia at multiple sites (Fig 5C). These results suggest that N-glycosylation is important for accumulating conidium storage reserves.

## N-glycosylation coordinates different glycosylation pathways during conidium and appressorium formation

We identified several proteins involved in various glycosylation pathways, including the N-glycosylation pathway, O-glycosylation pathway and GPI anchor pathway, that were modified by N-glycosylation. The N-glycosylation levels increased during conidium and appressorium formation (Fig 6).

Among the N-linked glycoprotein biosynthesis pathway, the N-linked glycan synthesis starts in the ER and continues into the Golgi apparatus. At least six proteins involved in this pathway are N-glycosylated themselves (Fig 6A). STT3, OST1 and WBP1 are subunits of the oligosaccharyltransferase complex of the ER lumen, which catalyze the linking of N-glycans onto newly synthesized proteins (Fig 6A and 6C). MNN10 and MNN9 are two alpha-1,6-mannosyltransferases, subunits of a Golgi-localized complex involved in glycan biosynthesis. MNN2 is an alpha-1,2-mannosyltransferase that is responsible for adding the first alpha-1,2-linked mannose to form the branches on the mannan backbone of oligosaccharides and was found to be localized into an early Golgi compartment. Interestingly, these proteins perform the later steps of the N-glycosylation pathway, indicating N-glycosylation has a feedback regulatory role in this part of the pathway.

GPI is a glycolipid which can be attached to the C-terminus of a protein. GPI plays a key role in numerous biological processes by targeting proteins on the cell wall or plasma membrane and modifying the anchors. Several key components of the GPI anchor pathway, PIG-N, PIG-K and PIG-T, were identified as N-glycosylated proteins (Fig 6B and 6C). PIG-N contains four N-glycosites, three of which (N145, N208 and N256) are highly N-glycosylated at the conidium and appressorium (Fig 6C). PIG-T contains one glycosite at N41 that was also highly N-glycosylated at these developmental stages (Fig 6C).

O-linked glycosylation is another type of glycosylation, which adds N-acetyl-galactosamine to serine or threonine residues and uses secretion to form the extracellular matrix components. Two of the three key components in the O-glycosylation pathway, PMT2 and PMT4, were identified as N-glycosylation proteins (Fig 6C). PMT2 and PMT4 are protein O-mannosyltransferases, which transfers mannose residues from dolichyl phosphate-D-mannose to protein serine/threonine residues in the ER membrane. Both PMT2 and PMT4 were proved to be important for infection of *M. oryzae* [39–40].

Collectively, N-glycosylation can modify key proteins in all the N-glycosylation, O-glycosylation and GPI anchor pathways, indicating a close crosstalk between these cellular processes.

## Most proteins involved in ERQC were highly N-glycosylated in the conidium and appressorium

In eukaryotic cells, the endoplasmic reticulum (ER) serves as a protein-folding factory where elaborate quality and quantity control systems monitor the efficient and accurate production of secretory proteins. The ER uses a quality control (ERQC) system to facilitate folding of secretory proteins, and eliminates misfolded proteins through the ER-associated degradation (ERAD) system or autophagic degradation system. Surprisingly, we found that near all

**A**

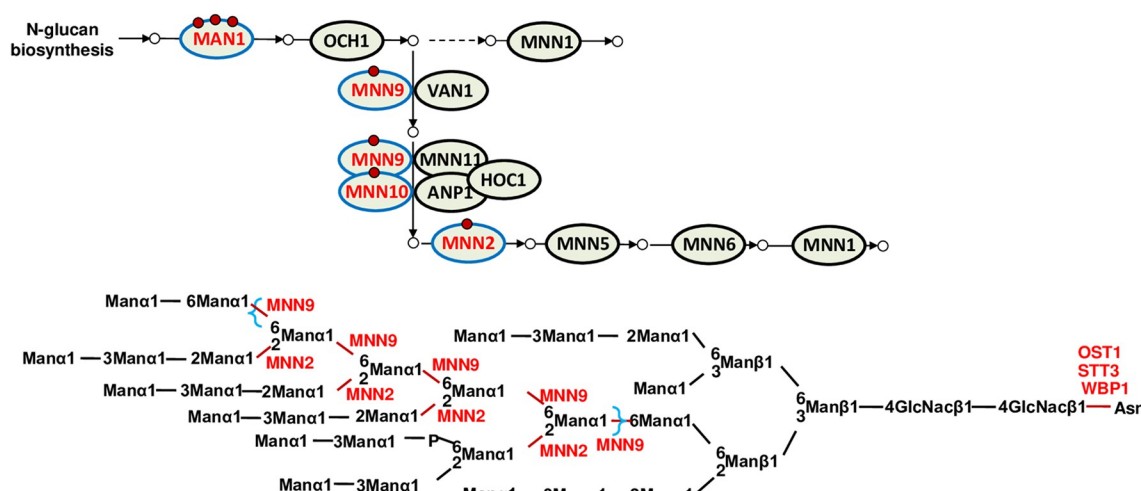

**B**

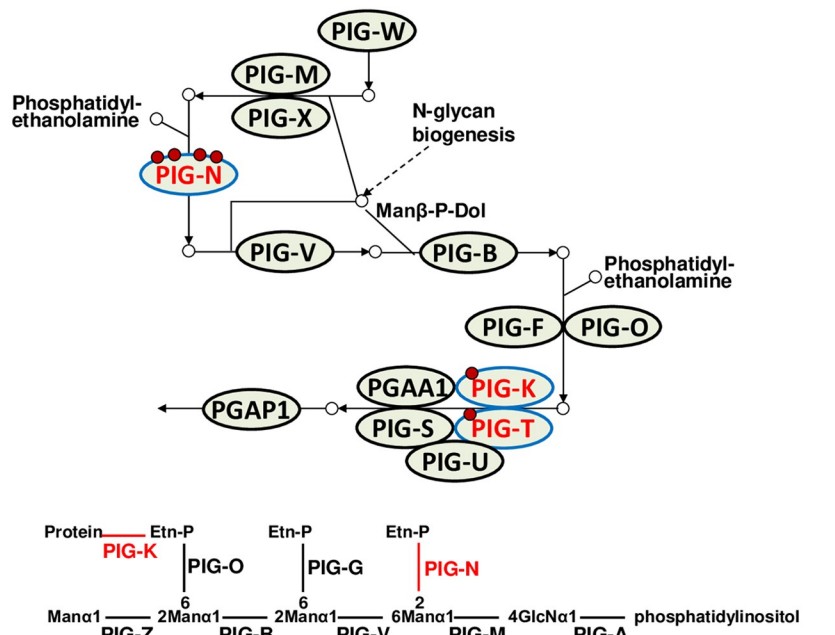

**C**

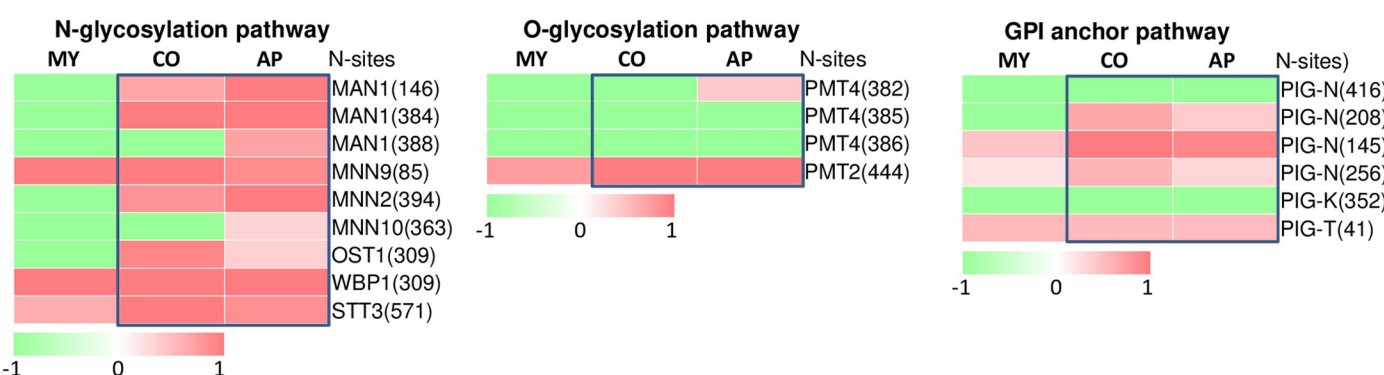

**Fig 6. Glycosylation pathways are mediated by N-glycosylation.** (A) N-glycan biosynthesis and (B) GPI anchor biosynthesis. (C) Changes in N-glycosylation protein levels in three glycosylation pathways at different stages. N-glycosylated proteins were marked in red font; N-glycosite was indicated by red ring. The colored bar represents the scale for the log10 fold change in expression from green (-2) to red (2), white is set as 0. MY, mycelia; CO, conidia; AP, appressoria.

components of the ERQC system are N-glycosylated (Fig 7), including Sec63, Kar2, Scj1, STT3, OST1, WBP1, Gls1, Gls2, GTB1, CNX1, Mns1, PDI1 and KRE5.

We found that N-glycosylation affects every step of the ERQC pathway (Fig 7A). Newly synthesized membrane and secretion-bound proteins enter the ER in an unfolded state through a translocon channel [41], of which the N-glycosylated protein Sec63 is an essential subunit. As the newly synthesized proteins enter the ER lumen, chaperones immediately engage with the nascent polypeptides to facilitate translocation and protein folding; these chaperones include the Hsp70 family member Kar2 and the DnaJ class co-chaperone Scj1, both are targets of N-glycosylation. Oligosaccharides are transferred to the proteins by oligosaccharyltransferases containing subunits STT3, OST1 and WBP1, all highly N-glycosylated at the conidium and appressorium stages (Fig 7B). Pre-Golgi processing of the nascent N-glyco-proteins involves the elimination of specific Glc and Man subunits by four conserved enzymes (Gls1, Gls2, GTB1, and MNS1), which are also highly N-glycosylated at the conidia or appressoria (Fig 7B). The processing is linked to the calnexin cycle, the main checkpoint for evaluating correct N-glycoprotein folding [42], where another highly N-glycosylated protein, the calnexin CNX1, verifies the presence of N-glycoproteins harboring the oligosaccharide core NAcGlc$_2$Man$_9$Glc [43]. Permanently misfolded proteins are transported to the proteasome for degradation [44], a process mediated by the N-glycosylated proteins PDI1, glucosyltransferase KRE5, and mannosidase Mns1 (Fig 7A).

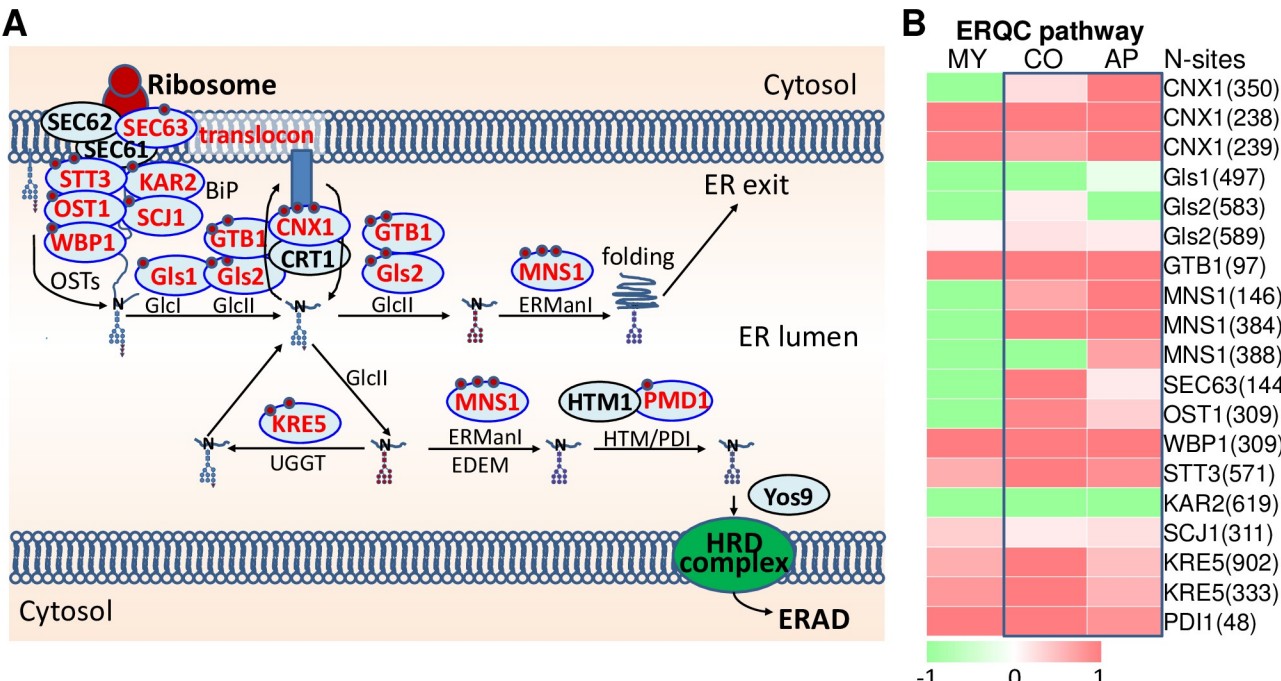

**Fig 7. The ERQC system is regulated by N-glycosylation.** (A) Schematic diagram of the ERQC pathway. N-glycosylated proteins were marked in red font; N-glycosites was indicated by red ring. (B) Changes in ERQC pathway N-glycosylation protein levels in three glycosylation pathways. The colored bar represents the scale for the log10 fold change in expression from green (-2) to red (2), white is set as 0. MY, mycelia; CO, conidia; AP, appressoria.

## N-glycosylation regulates the ERQC system for fungal development and infection

CNX1, Gls1, Gls2, and GTB1 were chosen to understand the N-glycosylation of the ERQC pathway proteins. We generated single gene deletion mutants for Gls1, Gls2, GTB1 and CNX1, and performed complementation by transforming single gene's coding region into its deletion mutant. All of corresponding complementation strains recovered their phenotypic defects (S2 Fig). Loss of these genes produced slight defects in colony growth (Fig 8A and 8B). Mutants of these genes showed no more than 50% conidiation capacity compared with the wild-type strain (Fig 8C). Microscopic observation showed that these mutants formed sparse conidiophores with fewer conidia (Fig 8D). To determine the roles of ERQC pathway proteins in virulence, Δgls1, Δgls2, Δgtb1 and Δcnx1 were inoculated onto host leaves. Interestingly, virulence of all the mutants was significantly reduced (Fig 7E). The mutants exhibited normal appressorium formation during infection (Fig 7F), but Δgls1, Δgls2 and Δcnx1 exhibited growth arrest inside the plant tissues just after appressorium penetration, and Δgtb1 had slowed invasive hyphal growth (Fig 8G and 8H). This indicated that Δgtb1 can complete the early stages of plant interaction after appressorium penetration, but is unable to efficiently spread fungi

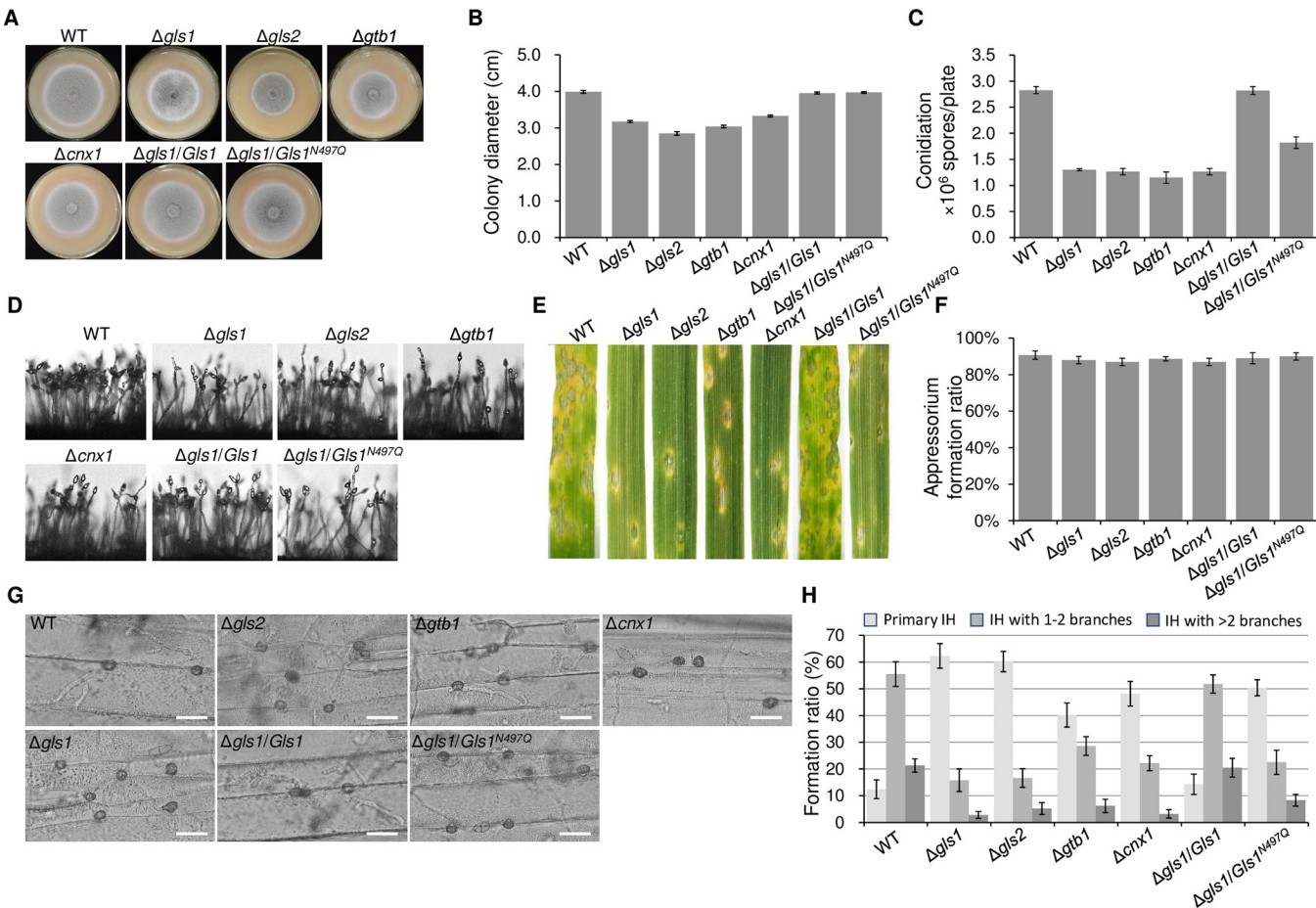

**Fig 8. Functional analysis of the N-glycosylated components of the ERQC pathway.** (A) Colony morphology of the wild-type strain P131, ERQC pathway gene deletion mutants and complementary strains of *Gls1* on oatmeal agar (OTA) plates. (B) Colony diameters of different strains. (C) Conidiation of different strains. (D) Conidiophore morphologies formed by different strains. (E) Lesions formed on barley leaves by different strains at 5 d post-inoculation (dpi). (F) Appressorium formation by different strains at 12 dpi. (G) Invasive hyphae (IH) formed by the same set of strains in barley epidermal cells at 24 h post-inoculation (hpi). Bar, 20 μm. (H) Formed ratio of IH in barley epidermal cells at 24 hpi.

inside plant tissues. Thus, ERQC is essential for establishing the biotrophic growth during infection of *M. oryzae*.

To verify regulation of the ERQC system by N-glycosylation, Gls1 was chosen as a target for validation of N-glycosylation. We constructed a GFP fusion construct of *Gls1* and transformed it into the Δ*gls1* or Δ*alg3* deletion mutant. ALG3 is one of the key components involved in N-glycan biosynthesis [13]. In mycelia, we failed to detect any difference in protein size of GFP: Gls1 between the Δ*gls1* mutant and the Δ*alg3* mutant, indicating that N-glycosylation was not frequent enough to be detected by western blot in this tissue (Fig 9A). However, in the appressorium, a protein band larger than GFP:Gls1 fusion protein in the wild-type strain was detected, while a protein band corresponding to GFP:Gls1 itself was detected in the Δ*alg3* mutant. When we used Endo H, a N-glycanase, to treat the wild-type proteins, the larger protein was disappeared and the GFP:Gls1 protein band was detected (Fig 9A). This experiment showed that Gls1 is N-glycosylated in the appressorium.

We simultaneously validated the N-glycosite of Gls1 identified in this study (Fig 7B). The GFP:Gls1 fusion construct with or without the glycosite mutation N497Q (Asn at 497aa was mutated as Gln) driven by the native promoter was transformed into the Δ*gls1* mutant, respectively. In appressoria, western blot analysis showed that the N497Q mutation resulted in a GFP: Gls1 fusion protein band represent the size of the unglycosylated protein, rather than the larger glycosylated band (Fig 9A), confirming that N497 is a functional N-glycosite. Functional analyses revealed that the Gls1$^{N497Q}$ mutation resulted in significant reduction of virulence and a block in the infection process (Fig 8E–8H), indicating that N-glycosylation is important for functions of Gls1. Interestingly, the Gls1$^{N497Q}$ mutation did not affect colony growth, and only partially affected conidiation (Fig 8A–8D). Subcellular localization analysis demonstrated that the GFP:Gls1 co-localizes with RFP-HDEL protein (Fig 9B–9D), which contains a ER-retention signal [45]. Therefore, the Gls1 was located in the ER at different developmental stages. The GFP:Gls1$^{N497Q}$ protein co-localized with the fluorescent vacuole marker dye CMAC (7-amino-4-chloromethylcoumarin, [46]) in the conidia and appressoria, and was hardly visible in invasive hyphae (Fig 9B–9D), demonstrating that N-glycosylation is required for the proper localization of ERQC proteins in ER. In sum, N-glycosylation is required for functions of ERQC proteins, probably through affecting their proper subcellular localization and protein stability.

## Discussion

Previous studies have found that protein N-glycosylation is essential during infection of plant pathogenic fungi [13–16], yet the systematic regulatory patterns of N-glycosylation are not well understood. In order to fully understand the roles of N-glycosylation in pathogenic fungi, we assessed the importance and modification levels of N-glycosylation at different developmental stages in *M. oryzae*. Quantitative measurements provided a molecular signature specific to glycoproteins for different developmental and infection stages. The established system for N-glycoproteome quantification in *M. oryzae* will be helpful in elucidating N-glycoproteome dynamics in fungi and other species.

Recently, N-glycoproteomics studies have been performed in the tissues of different organisms, including yeast [47–48], filamentous fungi [49–50], nematode [51], Drosophila [52], the plant *Arabidopsis thaliana* [53] and mammals [54–56]. Quantifying the N-glycoproteomes in the plant pathogenic fungus *F. graminearum* have revealed intensive glycosylation changes during exposure to fungicide: 774 sites in 406 proteins were identified, and the glycosylation level was found to be largely down-regulated upon fungicide treatment [50]. However, no study has focused on the quantitative N-glycosylation changes during normal development and infection processes of plant pathogenic fungi.

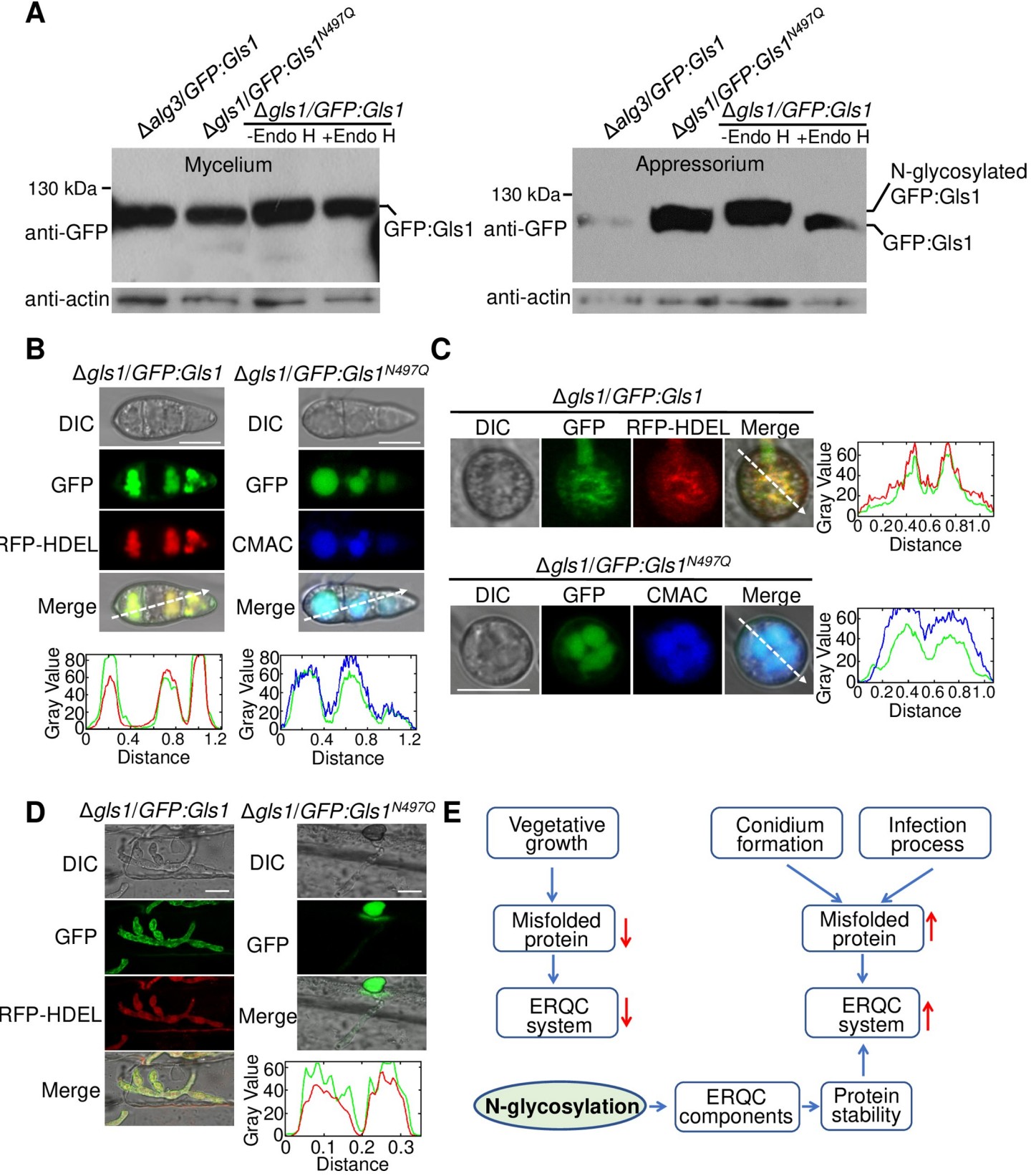

**Fig 9. Western blot analysis and subcellular localization of Gls1.** (A) Western blot of N-glycosylated ERQC proteins in mycelia and appressoria. Total proteins from extracts of the Δ*gls1* and the Δ*alg3* mutant expressing GFP-fused Gls1 were separated by SDS-PAGE and then subjected to western blot analysis with an anti-GFP antibody. Total proteins isolated from the transformants were treated with or without Endo H and detected with an anti-GFP antibody by immunoblot analysis. Anti-actin antibody was used to evaluate protein loading level. (B) Subcellular co-localization of GFP:Gls1/RFP:HDEL and GFP:Gls1$^{N497Q}$/CMAC in conidium. Bar, 10 μm. (C) Subcellular co-localization of GFP:Gls1/RFP:HDEL and GFP:Gls1$^{N497Q}$/CMAC in appressorium. Bar, 10 μm. (D) Subcellular localization of GFP:Gls1/RFP:HDEL and GFP:Gls1$^{N497Q}$ in invasive hyphae. Bar, 10 μm. For (B), (C) and (D), linescan graphs show fluorescence intensity in a transverse section of individual appressorium as the arrow indicated direction, line colors refer to corresponding fluorescence colors. (E) Proposed model of the N-glycosylation regulated ERQC system for development.

## N-glycosylation coordinates different cellular processes for development and host infection

Our high-throughput quantitative N-glycosylation proteomics study provides important novel insight into the development and infection process of *M. oryzae* (S3 Fig). We found that N-glycosylation may be important in coordinating different developments. Distinct proteins that corresponding specifically to the mycelium, conidium or appressorium stages were highly N-glycosylated. The N-glycosylated proteins at the vegetative growth stage were involved in nutrient utilization, cell wall biogenesis and the redox process; at the conidium formation stage, the N-glycosylated proteins were involved in glycogen storage, cell wall biogenesis, glycosylation (N-glycosylation, O-glycosylation and GPI anchor) and ER quality control; and at the appressorium formation stage, the N-glycosylated proteins were relative to glycogen utilization, lipid utilization, cell wall biogenesis, glycosylation pathways, and ER quality control (S3 Fig). We inferred that during invasive hyphal growth, proteins related to nutrient utilization, cell wall biogenesis, redox process and effector secretion could also be heavily N-glycosylated. This study demonstrated major changes in the N-glycosylation level of different stages. This novel approach enabled us to comprehensively analyze the dynamic remodeling processes of N-glycosylated proteins during *M. oryzae* differentiation and to provide unique insights into the underlying mechanisms.

Systems biology studies are useful to reveal infection mechanisms of the plant pathogenic fungi. Previous studies have focused on the transcript and proteomics profiling during appressorium formation of *M. oryzae* [57–59]. Global patterns of gene expression in appressorium were detected, highlighting the role of autophagy, sugar and lipid metabolism and melanin biosynthesis [59]. Phosphoproteomic profiling of *M. oryzae* during the appressorium formation stage provided insights into the metabolic regulation of conidial storage reserves and phospholipids, autophagy, actin dynamics and cell wall metabolism [58]. Here, we revealed the N-glycoproteomics profiling of *M. oryzae* during the appressorium formation stage. The proteins involved in glycogen utilization, lipid utilization and cell wall biogenesis were consistent with the transcript and proteomics profiling results.

However, it should be noted that some homogenous proteins of the known N-glycosylated proteins were not identified in this study. For example, *M. oryzae* EMP1 (MGG_00527) is a homogenous protein of Fusarium adhesive protein FEM1 [60, 61]. This protein is a putative GPI-anchored, N-glycosylated, extracellular matrix protein, which is preferentially expressed during appressorium stage and is required for appressorium formation and full virulence. Surprisingly, EMP1 was not detected in this study. We infer that this protein, as well as others, may be not abundant enough for detection, due to low expression, or missing on plastic surface during sample collection, or strong linkage with cell wall structure. On the other hand, even we used three different lectins (ConA, WGA and RCA120) for enrichment, there could be still some N-glycosylated proteins can't be enriched. In fact, as predicted by a bioinformatic tool NetNGlyc 1.0 (http://www.cbs.dtu.dk/services/NetNGlyc/), more than 50% of the total *M. oryzae* proteins contain predicted N-glycosylation site(s), and only a small proportion of them were identified in this study. Therefore, more efficient enrichment methods need to be developed for identifying more N-glycosylated proteins.

## N-glycosylation modifies proteins of different sugar-related biosynthesis pathways

We found that many proteins specific for ER or Golgi localization are found in the N-glyco-proteome. Proteins involved in the N-glycosylation pathway itself, such as the OST subunits Ost1, Wbp1, and Stt3, as well as some Golgi mannosyltransferases involved in N-glycan processing (MAN1, MNN9, MNN2, and MNN10), were found to be N-glycosylated. O-glycosylation is initially catalyzed by evolutionarily conserved protein O-mannosyltransferases (PMTs). In this study, two of the three key members of PMTs were identified as N-glycosylated proteins. Three key components of the GPI synthesis pathway (PIG-N, PIG-T, PIG-K) were also identified as N-glycosylated proteins. Therefore, all three types of glycosylation pathways could be regulated by N-glycosylation. Coincidently, previous studies on the O-glycoproteome in yeast also observed an interaction between O- and N-glycosylation [62]. Therefore, understanding the interactions between the different glycosylation synthesis pathways may help decipher the undoubtedly crucial role of protein glycosylation.

## The ER quality control system is extensively regulated by N-glycosylation

N-glycosylation of ERQC proteins provides new insight into host infection. It has been reported that ERQC plays key roles in infection of *Ustilago maydis* [14]. In *U. maydis*, glucosidase I Gls1 is required for the initial stages of infection following appressorium penetration, and glucosidase II β-subunit Gas2 (GTB1) is required for efficient fungal spreading inside infected tissues. Thus, the ERQC is important for establishing the initial biotrophic state in the plant and allows subsequent colonization of *U. maydis*.

In this study, we identified most of the ERQC key components were N-glycosylated target proteins in *M. oryzae*. We subsequently found that these ERQC components are required for host penetration and invasive hyphal growth in *M. oryzae*. More importantly, we proved that the N-site of Gls1 can regulate its function by affecting its subcellular localization in ER and protein stability, revealing roles of N-glycosylation in protein quality control and the infection process. As the N-site mutation of Gls1 did not affect normal colony growth but severely affected conidium formation and the infection process, we inferred that the ERQC system may be more important for conidiation and the infection processes when N-glycosylation is increased. N-glycosylation may keep protein stability of the ERQC components and help to stabilize the ERQC system (Fig 9E). This regulatory infection mechanism is a novel discovery that has not been previously demonstrated in the plant pathogenic fungi.

In summary, our study shows that N-glycosylation plays critical roles in the development and infection process of *M. oryzae* through a quantitative N-glycoproteome analysis. The results from this study would increase our understanding of the *M. oryzae* infection mechanism. The large pool of N-glycosylation targets identified here will serve as a valuable resource for further understanding how the N-glycosylation mediates fungal infection process, which may lead to developing novel fungal disease control strategies.

## Materials and methods

### Strains and culture conditions

The *M. oryzae* strain P131 was used as the wild type [13]. All of the wild-type strain and transformants used in this study (S5 Table) were grown on Oatmeal Tomato Agar (OTA) plates at 28°C. Mycelia were incubated at 28°C in liquid culture medium on a rotary shaker (180 rpm) for 36 h. Colony growth and conidiation were performed as described previously [13]. Conidia harvested from 7-day-old OTA cultures were used to test virulence and to observe the

infection process. Appressorium formation was performed by dropping a conidial suspension ($1 \times 10^5$ conidia/mL) onto a hydrophobic coverslip and incubated in a dark, moist chamber at 28˚C.

## Staining assays

N-glycoproteins during different developmental stages of *M. oryzae* were stained with 0.1 mg/mL FITC conjugated concanavalin A (Sigma-Aldrich, USA) for 30 min before analysis. To detect the GPI-anchored proteins on the cell wall, 50 nM fluorochrome (Alexa 488)–labeled inactivated aerolysin (FLAER) (Pinewood Scientific Services, Victoria, Canada) was used to stain different samples of *M. oryzae* for 30 min before observation. For invasive hyphae staining, the conidia suspension was dropped onto the barley leaves and incubated with full humidity, then ConA-FITC or FLAER was added into the conidia droplets at 30 h for staining and observed after 30 min. Both ConA-FITC and FLAER staining samples were observed under a confocal laser scanning microscope (Leica SP8, Leica Microsystems, Germany). The fluorophore was excited at 488 nm, and the emission was detected at 530 nm (530±10 nm), with a same gain value of 800. To observe the cell lengths of hyphal tips, 10 μg/mL clacofluor white (CFW) (Sigma-Aldrich, USA) was used to stain hyphal cell walls and septa for 10 min in the dark, and the hyphal tips were observed under a fluorescence microscope (Nikon Ni90 microscope, Japan) after being rinsed with PBS buffer. The vacuoles of different tissues were stained with CMAC (7-amino-4-chloromethylcoumarin) (Sigma-Aldrich, USA) for 10 min in the dark, and observed under a confocal laser scanning microscope (Leica SP8, Leica Microsystems, Germany).

## Infection assays

To test the virulence of different fungal strains, 1-week-old barley leaves (*Hordeum vulgare* cv. E9) were sprayed with conidial suspensions ($1 \times 10^5$ conidia/mL) in 0.025% Tween 20. The inoculated plants were incubated at 28˚C with full humidity, and the disease lesions were observed at 5 dpi. The infection process was identified in the host cells by inoculating the lower barley leaves with a conidia suspension ($1 \times 10^5$ conidia/mL) and incubated in a dark, moist chamber at 28˚C. The lower barley epidermis was torn down to observe the infection processes at 24 hpi. The effects of 5 μg/mL Tunicamycin (Sigma-Aldrich, USA) or 10 mM Dithiothreitol (DTT, Sigma-Aldrich, USA) at different developmental stages were performed by different treatment methods. The mycelia were treated with either TM or DTT after 24 h incubation in liquid complete medium (CM), and then added with Tunicamycin or DTT, continued to incubate for 12 h. At last, the mycelia dry weight was evaluated and the mycelial morphology was observed by CFW staining. The conidia were produced by spreading mycelial fragments onto the OTA plates containing TM or DTT for conidiation, and the conidiophores were observed after 12h. The effect on appressorium was performed by adding Tunicamycin into the conidia droplets on hydrophobic surface, and the appressorium formation was observed at 12 hpi. Effect on invasive hyphae was performed by adding Tunicamycin into the conidia droplets on barley leaves at 18 hpi, and the invasive hyphal growth was observed at 24 hpi.

## Preparation of protein samples

To perform quantitative N-glycoproteomics analysis, the mycelia and conidia were harvested as mentioned above and were stored at −80˚C before processing. For appressoria collection, high concentration of the conidia suspension ($2\times10^6$ conidia mL$^{-1}$) were sprayed onto a 21 cm×29.7 cm sized hydrophobic plastic surface and incubated in dark, moist plates at 28˚C.

After 6 h, 12 h or 24 h, the appressoria were collected by scraping plastics surface and filtered with six layers lens wiping paper. Lysis buffer containing 45 mL TCA:acetone (1:9) and 65 mM DTT was added to the frozen samples, followed by refrigerated centrifugation, washing, boiling and sonication. The total protein was quantified using the BCA method.

### Protein trypsin digestion

Total proteins were processed according to the "filter-aided sample preparation" (FASP) method as reported. A total of 1.2 mg of protein for each sample was dissolved in a solution containing 8 M urea, 0.1 M $NH_4HCO_3$ and Tris-HCl pH 8.0 in a 10-kDa ultrafiltration tube. The samples were subsequently alkylated for 30 min with 50 mM iodoacetamide in the dark and then centrifuged for 10 min at 14000 g. Samples were diluted with UA buffer (8 M urea, 150 mM Tris-HCl, pH 8.0) and washed with 25 mM $NH_4HCO_3$ three times. The proteins were incubated with trypsin (proteins/trypsin, 50:1, w/w) at 37˚C for 18 h. The digestive reaction was stopped by adding formic acid, and the peptides were desalted on a solid phase extraction C18 cartridge (Waters, MA).

### Lectin enrichment and deglycosylation

After digestion, peptides were eluted in the lectin binding buffer (1 mM $CaCl_2$, 1 mM $MnCl_2$, 0.5 M NaCl, 20 mM Tris-HCl [pH 7.3]) and subsequently transferred to a 30 kDa filtration unit. Lectin mixture containing concanavalin A (ConA), wheat germ agglutinin (WGA), and *Ricinus communis* agglutinin (RCA120) was added to the top of the filters. The non-bound peptides were eluted, and the captured peptides were washed with the binding buffer after adding PNGase F. Deglycosylation was performed in $H_2^{18}O$ for 3 h at 37˚C and the deglycosylated peptides were eluted and analyzed.

### Tandem mass spectrometry analysis

The deglycosylated peptides were resuspended in 0.1% FA, then analyzed with LC−MS/MS. LC-MS/MS was performed using a Q-Exactive mass spectrometer coupled with an Easy nLC (Thermo Fisher Scientific, MA). Testing samples were loaded onto a 100 μm × 20 mm C18 precolumn (Thermo Fisher Scientific, CA) and chromatographic separation was performed on a 75 μm × 10 cm C18 nanocolumn. A high-performance liquid chromatography gradient was achieved using 0−55% buffer B (0.1% FA and 95% acetonitrile) and buffer A (0.1% FA) at a flow rate of 300 nL/min for more than 90 min. A Q-Exactive mass spectrometer was used for the analysis. The MS data was acquired with a precursor ion range of 300−1800 in positive ion mode. Resolution was set to 70 000 at m/z 200, the dynamic exclusion 25 s, maximum ion injection time 20 ms and automatic gain control target $3 \times 10^6$. Higher-energy collisional dissociation at a normalized collision energy of 27 eV was used to acquire the 10 most intense ions for MS2 scans with a resolution of 17500 at m/z 200 and 60 ms maximum IT.

### Data analysis

The label-free quantitative analysis of N-glycosylated peptides was performed using the MaxQuant software version 1.3.0.5 according to a previous study [63]. Tandem mass spectra were searched against the *M. oryzae* proteome database (http://fungi.ensembl.org/Magnaporthe_oryzae/Info/Index). Ten LC−MS/MS raw files were obtained from two replicates of five samples. The search parameters were set as follows: enzyme cleavage was trypsin; the maximum missed cleavage sites were two; variable modifications were oxidation on methionine and

deamidation ($^{18}$O) in asparagine; peptide mass tolerance was 10 ppm; false discovery rate (FDR) thresholds were 0.01.

## Bioinformatics analysis

The Gene Ontology (GO) annotation of the N-glycoproteome was performed using the Uni-Prot-GOA database (http://www.ebi.ac.uk/GOA/). The 'biological processes', 'molecular functions' and 'cellular components' categories were used to classify N-glycosylated proteins based on GO annotation. The Kyoto Encyclopedia of Genes and Genomes (KEGG) database was used to annotate the N-glycosylated protein pathways. KAAS, an online service tool, was used to annotate the KEGG database description of the N-glycosylated proteins (http://www.genome.jp/tools/kaas/). Resulting annotations were used to map the KEGG pathway database. The STRING database (https://string-db.org/) and Cytoscape software (version 3.4.0) were used to identify N-glycoprotein interaction networks [64–65]. The principal component analysis (PCA) was performed using the psych R-package with standard settings. The heatmap was created with the NMF R-package. Different samples were compared using the cor function of the stats R-package reporting a Pearson correlation used on complete observations only. To create groups of same expression profiles, the fold change was calculated to a specific sample. Any fold change larger than 2 with $p<0.05$ (Welch t-test) was reported as significantly up- or down-regulated.

## Western blotting

Mycelia samples were incubated and collected from liquid CM medium cultures. Appressoria samples were collected from the hydrophobic surface sprayed by conidia with a high concentration ($2\times10^6$ spores/mL). To detect N-glycosylation level, total proteins of different samples were extracted and mixed with SDS-PAGE loading buffer, denatured at 95°C for 5 min and then subjected to 12% SDS-PAGE for western blot analysis using ConA-HRP (1:10,000) as an antibody. To generate GFP-fused constructs, the promoter and coding regions of Gls1 was amplified and fused with GFP, then cloned into pKN(S6 Table). In the resulting vectors, the GFP fusion constructs were expressed under the control of the native promoter [48]. These vectors were transformed to protoplasts of P131, the Δ*alg3* mutant or the Δ*gls1* mutant to obtain the GFP-tagged strains. Total proteins were extracted to perform western blot using an anti-GFP antibody (1:5,000; Abmart) as the primary antibody. For N-glycosite mutation, the GFP:Gls1$^{N497Q}$ fusion construct was constructed and transformed into the Δ*gls1* mutant.

## Supporting information

**S1 Fig. Expression profile of the N-glycosylation pathway genes during development.** The phase specific expression of these genes was quantified by quantitative real-time PCR with synthesis of cDNA from each sample including mycelia, conidia, germ tubes, appressoria and invasive hyphae at indicated time points. Relative abundance was normalized by *MoTub1*. Three independent biological experiments with three replicates in each were performed. MY: mycelia; CO: conidia; AP: appressoria; IH: invasive hyphae.
(TIF)

**S2 Fig. Phenotypes of the complement strains.** (A) Colony diameters of the wild-type (WT) and different complement strains. (B) Conidiation of the WT and different complement strains. (C) Invasive hyphae (IH) formed by the same set of strains in barley epidermal cells at 24 h post-inoculation (hpi). Bar, 20 μm. (D) Formed ratio of IH in barley epidermal cells at 24

hpi.
(TIF)

**S3 Fig. Diagram representation of N-glycosylation functions at *M. oryzae* different developmental stages.**
(TIF)

**S1 Table. List of all identified N-glycosites in *M. oryzae*.**
(XLSX)

**S2 Table. List of all identified N-proteins in *M. oryzae*.**
(XLSX)

**S3 Table. Differentially expressed N-glycosylation sites during different developmental stages.**
(XLSX)

**S4 Table. Differentially expressed N-glycosylation sites for different processes.**
(XLSX)

**S5 Table. Fungal strains used in this study.**
(XLSX)

**S6 Table. Plasmids used in this study.**
(XLSX)

## Acknowledgments

We thank Dr. Lindsay Triplett in Connecticut Agricultural Experiment Station (CAES) of USA for her critical reading of the manuscript. We also thank Shanghai Applied Protein Technology, Co. for providing technical support.

## Author Contributions

**Conceptualization:** Xiao-Lin Chen, Wende Liu.

**Data curation:** Xiao-Lin Chen, Wende Liu.

**Formal analysis:** Xiao-Lin Chen, Bozeng Tang, Wende Liu.

**Funding acquisition:** Xiao-Lin Chen, Wende Liu.

**Investigation:** Xiao-Lin Chen, Caiyun Liu, Zhiyong Ren, Wende Liu.

**Methodology:** Xiao-Lin Chen, Caiyun Liu, Zhiyong Ren.

**Project administration:** Xiao-Lin Chen, Wende Liu.

**Resources:** Wende Liu.

**Software:** Xiao-Lin Chen, Bozeng Tang.

**Supervision:** Guo-Liang Wang, Wende Liu.

**Validation:** Xiao-Lin Chen, Wende Liu.

**Visualization:** Xiao-Lin Chen, Wende Liu.

**Writing – original draft:** Xiao-Lin Chen, Wende Liu.

**Writing – review & editing:** Xiao-Lin Chen, Guo-Liang Wang, Wende Liu.

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
