## [Decision Letter · Decision Letter 0]

9 Dec 2019

Dear Dr. Liu:

Thank you very much for submitting your manuscript "Quantitative proteomics analysis reveals important roles of N-glycosylation on ER quality control system for development and pathogenesis in Magnaporthe oryzae" (PPATHOGENS-D-19-01853) for review by PLOS Pathogens. Your manuscript was fully evaluated at the editorial level and by independent peer reviewers. The reviewers appreciated the attention to an important topic but identified some aspects of the manuscript that should be improved.

We therefore ask you to modify the manuscript according to the review recommendations before we can consider your manuscript for acceptance. Your revisions should address the specific points made by each reviewer.

(1) A letter containing a detailed list of your responses to the review comments and a description of the changes you have made in the manuscript. Please note while forming your response, if your article is accepted, you may have the opportunity to make the peer review history publicly available. The record will include editor decision letters (with reviews) and your responses to reviewer comments. If eligible, we will contact you to opt in or out.

(2) Two versions of the manuscript: one with either highlights or tracked changes denoting where the text has been changed; the other a clean version (uploaded as the manuscript file).

We hope to receive your revised manuscript within 60 days or less. If you anticipate any delay in its return, we ask that you let us know the expected resubmission date by replying to this email.

[LINK]

Sincerely,

Jin-Rong Xu, PhD

Associate Editor

PLOS Pathogens

Bart Thomma

Section Editor

PLOS Pathogens

Kasturi Haldar

Editor-in-Chief

PLOS Pathogens

orcid.org/0000-0001-5065-158X

Grant McFadden

Editor-in-Chief

PLOS Pathogens

orcid.org/0000-0002-2556-3526

Reviewer's Responses to Questions

**Part I - Summary**

Reviewer #1: (No Response)

Reviewer #2: The manuscript defines the detectable set of N-linked glycosylated proteins from Magnaporthe oryzae mycelia, conidia, and germinated conidia that have produced appressoria. The function of N-glycosylation includes stabilizing protein structure and affecting other modifications, and its pattern may reflect the specialization of different cell types.

Here the authors have shown differential glycosylation patterns for the different cell types, that proteins involved in ER quality control are N-glycosylated, and that deletion of the ER quality control proteins leads to defective growth and development, and in growth in planta. One of these, gls-1 was assessed for the role of glycosylation and a change in the glycosylation at one site had little effect on growth of mycelia, a small effect on conidiation, but played a major role in growth in planta. This mutation appears to reduce stability of the protein so it may act as a knock down allele.

Results:

Biological importance of N-glycosylation at different development stages of M. oryzae

The first paragraph is weak. conA binds to glycoproteins, but also carbohydrates and glycolipids. Therefore staining of cell walls does not necessarily directly reflect glycoprotein level. If the authors were to provide evidence that carbohydrate and glycolipid levels are the same in cell walls of the different cell types, and the difference in staining is only due to glycoprotein binding, this would make sense. The exposure time for Fig 1A is not provided in the methods or figure legend. Presumably these images would all have the same exposure time based on the author’s use of the images to inform the reader about their interpretation of glycoprotein level. However, if the exposure times are the same, why is the conidium cell wall in the conidium image stained very well, but the conidial cell wall does not stain at all with conA in the appressorium image?

It is not stated how the sample for conA staining of the infected cell was prepared. Is this a live cell and conA can enter the rice cell to stain the fungal cell wall or was the plant wall permeabilized in some way? I assumed the in vitro images are all live cells to which conA was added, but the protocol may have been different for the in planta experiment?

This raises an interesting question. Spore tip mucilage provides a mechanism for surface attachment in spores. Whether spore tip mucilage is primarily carbohydrate or mixed carbohydrate and glycoprotein adhesive is not resolved. Surprisingly, spore tip mucilage is absent from image of the conidium in Fig 1A, why is that? Did conidia with spore tip mucilage have such a high level of fluorescence that the cell wall could not be visualized?

Would it be possible to extract the glycoprotein away from free carbohydrate to identity such glycoprotein(s) in spore tip mucilage? The authors are possibly thinking in this direction, so this work can lay the foundation for a future effort.

It seems to me that Fig 1B is where the authors can should start and make the argument that there is an overall lower level of glycoproteins relative to actin in mycelia. There are five cell types assessed in Fig 1A, in vitro mycelia, conidia, germ tube, appressorium, and hyphae in planta. Although Fig1A does not address the amount of glycoprotein in the cell wall, it does clearly show the fundamentally different nature of cell walls in different cell types. The germ tube and appressorium have much stronger staining then the conidium.

The statement that TM was used “to determine whether N-glycosylation is important for M. oryzae development and pathogenesis” is not very useful. We know a lot about glycosylation and the ER stress response. In other systems, with the dose used here, prolonged ER stress triggers programmed cell death. We don’t know from the methods or text how long TM treatment was except for the plant infection assay, where the treatment was just 6 hours long. Six hours is a very reasonable exposure time where one would expect ER stress induction and pro-survival signaling to be occurring. Beyond 24 hours, one might expect pro-death signaling to begin, hence, colony growth and other measures may be examining a different type of response than a shorter exposure. It seems much too simplistic to simply say this is a determination of whether N-glycosylation is important. We already know it is important. How is it important. The authors address this point later. But again, the effect of TM on colony growth may be a very different thing than how N-glycosylation is important for development and pathogenesis. The effect of the 6 hour TM exposure on infection is relevant, however, DTT also induces ER stress, but does not directly effect glycosylation. Therefore, a critical control experiment that could easily have been done is absent. Is it glycosylation or ER stress that has an impact on these observations?

Quantitative N-glycoproteomics profiling of different development stages in M. oryzae

The methods are described well except for the harvesting of the appressorium samples. Fig 2 shows the timing for when cultures were harvested but not the method. In the methods it is stated: “To perform quantitative N-glycoproteomics analysis, the mycelium, conidium and appressorium were harvested as mentioned above and were stored at −80°C before processing.”

It was frustrating that the harvesting method for appressoria was actually not mentioned above.

The finding of differential glycoprotein abundance in different tissues is consistent with Fig. 2B. However, there is a minor issue that bothers me. EMP1 was identified by homology to a Fusarium adhesive protein, and is a putative GPI-anchored, N-glycosylated, extracellular matrix protein expressed preferentially during appressorium development (Ahn et al. 2004. Mol Cells 17:166-173). EMP1 mutants have reduced appressorium formation and pathogenicity. Surprisingly, EMP1(MGG_00527) was not detected in this study. Perhaps the protein was not abundant enough, or for whatever reasons was not harvested (stuck to plastic) or unextractable (linked to cell wall). However, it would be nice to use this as an example in the discussion to alert the reader to potential limitations.

The following sections describe the findings of the N-glycoprotein analysis and will be very interesting to readers who are interested in the cell wall and ER function.

Sequence features of N-glycoproteome

Classification of identified N-glycoproteins

N-glycoprotein protein interaction networks

Comparing N-glycoproteins at different developmental stages of M. oryzae

The only issue here is that the time course of appressorium formation is described in Fig. 2 and mentioned a few times in the text but there is no real mention of analysis. Mycelia vs conidia vs appressoria (combined time course of appressorium formation) is discussed, but no discussion of 6 vs 12 vs 24 hours of conidiation ever appears. If there is no statistically significant difference between the 6, 12, and 24 hour time points, this is important for the reader to know. But I presume the authors expected there to be significant differences between these time points, so probably this should be discussed more than it is. The reader will be curious to know the authors’ explanation for the lack of differences between these.

N-glycosylation regulates different cellular processes in development

N-glycosylation coordinates different glycosylation pathways during conidium and appressorium formation

Most proteins involved in ERQC were highly N-glycosylated in the conidium and appressorium

The concluding sentence of this section the authors say that ERQC is regulated by N-glycosylation. Since the title of the next section says this, maybe don’t need this sentence here.

N-glycosylation regulates the ERQC system for fungal development and infection

It is an intriguing finding that there is differential glycosylation of proteins involved in ERQC during development. The glycosylation is required for stability and localization of the proteins. Is this differential glycosylation of ERQC components required for ERQC function or does it reflect the basal capacity for the response? Clearly ERQC must operate in all cell types. Is ER stress normally induced during appressorium formation? Does N-glycosylation regulate development or does development regulate N-glycosylation? The authors clearly show a relationship between development and ERQC and this is of interest to readers. Overall, the authors have done a good job in explaining the significance of the work.

DISCUSSION:

The question of N-glycosylation of proteins during infection is important. Presumably, during infection, the host environment is hostile and phytochemical activity, cell wall attack, reactive oxygen species, and other host strategies might induce stress along with the endogenous stress the fungus is undergoing via its rapid growth and deployment of pathogen strategies (chemical, protein, etc.). The pathogen is to be expected during infection.

The manuscript is well written with only a few grammatical issues. There are some statements that are rationalizations. For example, starting at line 438, the statement that ERQC works at a low level in mycelia and a higher level in conidiation and infection where higher protein synthesis leads to unfolded proteins, inducing N-glycosylation of the proteins of the N-glycosylation pathway to allow them to more effectively glycosylate proteins to assist in the ERQC. Is that correct?

The statement on line 445. It is best left up to the reader to decide if the study provides deep insight. Some might have wanted to see the glycosylation patterns of proteins during growth in planta. The authors say that there is too small an amount of fungal material, however, I’m not sure this can be accepted at face value. Was an attempt made to detect the fungal N-glycosylated proteins in heavily infected leaf sheaths? It would be a deep insight to know if the fungus, after successfully overcoming the plant defense, is growing in an unstressed (like mycelia) or is always stressed (like appressoria) while growing in the plant.

Again, it would be nice to have a discussion of known or expected glycoproteins and some discussion of the limitations of the study. How many glycoproteins might one expect to find in the N-glycoproteome (based on bioinformatics?) and how many were identified. Suggestions for improving the techniques.

OVERALL this is an important study. The entry into the paper should emphasize Fig 1B and carefully interpret the meaning of the conA studies of Fig. 1A.

The emphasis on using the word regulation in describing the relationship between N-glycosylation and cell types should be more open to alternative explanations, for example N-glycosylation pathways might be regulated by signaling pathways that also regulate development. Both G-protein and cAMP pathway activation contribute to appressorium development. Does activation of those signaling pathways affect glycosylation levels? If the term regulation is to be used the authors should more clearly define what they mean by regulation.

**Part II – Major Issues: Key Experiments Required for Acceptance**

Reviewer #1: (No Response)

Reviewer #2: none

**Part III – Minor Issues: Editorial and Data Presentation Modifications**

Reviewer #1: The manuscript by Chen et al reports on quantitative N-glycoproteomics analysis to reveal systematic roles of N-glycosylation during fungal infection. The authors uncovered overall regulatory mechanisms of N-glycosylation at different development and infection stages of M. oryzae, and identified the endoplasmic reticulum quality control (ERQC) system were highly N-glycosylated, which is important for infection. They also proved N-glycosylation of the N497 site in ERQC component Gls1 was important for invasive growth. To my knowledge, this is the first time to uncover global roles of N-glycosylation during development and infection in the plant pathogenic fungi. Overall, this manuscript contains massive data and most of their conclusions are well supported. Here I list several points for consideration or suggestions for improving the manuscript.

Minor comments:

- For Fig. 8B, C and D, co-localization would be benefitted by the fluorescence intensity profiles.

- Phenotypic analyses to the complementary strains of the ERQC genes should be provided as a supplemental data.

- There are some spelling or grammatical errors can be found in the text, the authors need to comprehensively re-read the whole text to revise them. Some of these errors have been shown as follows.

- Page 2, line 27, ‘affect’ to ‘affected’.

- Page 3, line 39, ‘system’ to ‘systemic’.

- Page 4, line 54, ‘modifying’ to ‘altering’.

- Page 5, line 77, ‘had’ is not necessary.

- Page 5, line 91-95, two ‘which’ are used in one sentence, please refraise.

- Page 6, line 104, ‘were exist’ to ‘exist’.

- Page 6, line 110, ‘due to difficulty in separating’ to ‘due to difficulty separating’.

- Page 7, line 124, ‘in untreated’ to ‘without treatment’.

- Page 8, line 178, ‘on’ to ‘with’.

- Page 10, line 257, ‘involve’ to ‘involved’.

- Page 13, line 284, ‘to localize into’ to ‘to be localized into’.

- Page 14, line 299, ‘proved’ to ‘were proved’.

- Page 14, line 316, ‘both targets of’ to ‘both are targets of’.

- Page 17, line 391, ‘that corresponded’ to ‘corresponding’.

- Page 19, line 444, ‘stable’ to ‘stabilize’.

- Page 22, line 507, ‘peptides eluted and analyzed’ to ‘peptides were eluted and analyzed’.

Reviewer #2: This is included in the summary above

PLOS authors have the option to publish the peer review history of their article (what does this mean?). If published, this will include your full peer review and any attached files.

Reviewer #1: No

Reviewer #2: No

---

## [Decision Letter · Decision Letter 1]

27 Jan 2020

Dear Dr. Liu,

We are pleased to inform you that your manuscript 'Quantitative proteomics analysis reveals important roles of N-glycosylation on ER quality control system for development and pathogenesis in Magnaporthe oryzae' has been provisionally accepted for publication in PLOS Pathogens.

Before your manuscript can be formally accepted you will need to complete some formatting changes, which you will receive in a follow up email. A member of our team will be in touch within two working days with a set of requests.

Best regards,

Jin-Rong Xu, PhD

Associate Editor

PLOS Pathogens

Bart Thomma

Section Editor

PLOS Pathogens

Kasturi Haldar

Editor-in-Chief

PLOS Pathogens

orcid.org/0000-0001-5065-158X

Michael Malim

Editor-in-Chief

PLOS Pathogens

orcid.org/0000-0002-7699-2064

Reviewer Comments (if any, and for reference):

Reviewer's Responses to Questions

**Part I - Summary**

Reviewer #1: The authors have revised and improved the manuscript according to the comments. I think it is now suitable for publication in PLOS Pathogens.

Reviewer #2: The revisions to the manuscripts were very thorough and addressed the reviewer criticisms adequately.

**Part II – Major Issues: Key Experiments Required for Acceptance**

Reviewer #1: (No Response)

Reviewer #2: The revisions to the manuscripts were very thorough and addressed the reviewer criticisms adequately.

**Part III – Minor Issues: Editorial and Data Presentation Modifications**

Reviewer #1: (No Response)

Reviewer #2: None

PLOS authors have the option to publish the peer review history of their article (what does this mean?). If published, this will include your full peer review and any attached files.

Reviewer #1: No

Reviewer #2: No

---

## [Editor Report · Acceptance letter]

18 Feb 2020

Dear Dr. Liu,

We are delighted to inform you that your manuscript, "Quantitative proteomics analysis reveals important roles of N-glycosylation on ER quality control system for development and pathogenesis in Magnaporthe oryzae," has been formally accepted for publication in PLOS Pathogens.

Best regards,

Kasturi Haldar

Editor-in-Chief

PLOS Pathogens

orcid.org/0000-0001-5065-158X

Michael Malim

Editor-in-Chief

PLOS Pathogens

orcid.org/0000-0002-7699-2064